# Multi-omic underpinnings of epigenetic aging and human longevity

Lucas A. Mavromatis [1,3], Daniel B. Rosoff[1,2,3], Andrew S. Bell[1], Jeesun Jung [1], Josephin Wagner[1] & Falk W. Lohoff [1] ✉

Biological aging is accompanied by increasing morbidity, mortality, and healthcare costs; however, its molecular mechanisms are poorly understood. Here, we use multi-omic methods to integrate genomic, transcriptomic, and metabolomic data and identify biological associations with four measures of epigenetic age acceleration and a human longevity phenotype comprising healthspan, lifespan, and exceptional longevity (multivariate longevity). Using transcriptomic imputation, fine-mapping, and conditional analysis, we identify 22 high confidence associations with epigenetic age acceleration and seven with multivariate longevity. *FLOT1*, *KPNA4*, and *TMX2* are novel, high confidence genes associated with epigenetic age acceleration. In parallel, cis-instrument Mendelian randomization of the druggable genome associates *TPMT* and *NHLRC1* with epigenetic aging, supporting transcriptomic imputation findings. Metabolomics Mendelian randomization identifies a negative effect of non-high-density lipoprotein cholesterol and associated lipoproteins on multivariate longevity, but not epigenetic age acceleration. Finally, cell-type enrichment analysis implicates immune cells and precursors in epigenetic age acceleration and, more modestly, multivariate longevity. Follow-up Mendelian randomization of immune cell traits suggests lymphocyte subpopulations and lymphocytic surface molecules affect multivariate longevity and epigenetic age acceleration. Our results highlight druggable targets and biological pathways involved in aging and facilitate multi-omic comparisons of epigenetic clocks and human longevity.

Aging is often accompanied by a loss of independence, disability, and the onset of diseases like cancer, cardiovascular disease, and neurodegenerative diseases which cumulatively represent the leading cause of death in developed nations[1]. Traditionally, medical interventions seek to delay the onset of, cure, or treat the symptoms of individual age-related diseases. However, recently, describing and pharmacologically targeting the biological processes linking age to functional and health decline has received attention as an alternative strategy for increasing healthy years lived[2,3]. This strategy could offer health and economic gains that substantially outweigh those achieved by targeting specific diseases[3]. However, the feasibility of slowing biological aging is currently limited. Long, expensive clinical trials would be required to identify anti-aging interventions that increase life expectancy in healthy individuals[4]. Moreover, while promising drug candidates for specific diseases can be prioritized and even approved by regulators if they alter well-validated disease biomarkers (e.g., low-density lipoprotein cholesterol for cardiovascular disease), biomarkers for biological aging are poorly understood and validated.

[1]Section on Clinical Genomics and Experimental Therapeutics, National Institute on Alcohol Abuse and Alcoholism, National Institutes of Health, Bethesda, MD, USA. [2]NIH-Oxford-Cambridge Scholars Program, University of Oxford, Oxford, UK. [3]These authors contributed equally: Lucas A. Mavromatis, Daniel B. Rosoff. ✉e-mail: falk.lohoff@nih.gov

Epigenetic clocks, which incorporate data about the methylation of CpG sites across the human genome into weighted linear equations to predict chronological age and/or age-related endpoints[5], are generally considered to be the most promising biomarker for biological aging[6], and there have been several recent clinical trials aiming to slow epigenetic clocks with pharmacological and lifestyle interventions[7,8]. These clocks show strong correlations with chronological age and other aging-related phenotypes[9]. Additionally, epigenetic clocks are influenced by genetic factors and have heritability estimates ranging from 0.10 (single nucleotide polymorphism (SNP)-based heritability of GrimAge acceleration) to 0.43 (pedigree-based heritability of accelerated Horvath and Hannum clocks)[10,11]. For some individuals, epigenetic age outpaces chronological age in what is referred to as epigenetic age acceleration (EAA). EAA is associated with a number of health conditions and age-related diseases, including substance use behaviors[12–14], atherosclerosis[15], cancer[16], and mortality[17]. Given the potential of EAA as an aging biomarker and the large benefits that could be achieved through interventions that slow biological aging processes, it is crucial to describe the biological correlates of EAA, identify targetable biomolecules that modify EAA, and assess the biological similarities and differences between EAA and clinically relevant aging phenotypes like healthspan, lifespan, and exceptional longevity.

In this study, we use multi-omic methods to comparatively analyze EAA and a multivariate, longevity-related phenotype (referred to hereafter as multivariate longevity) comprising parental lifespan, heathspan, and exceptional longevity (Fig. 1 displays a study overview). We leverage data from genome-wide association studies (GWASs) of four epigenetic clocks[10] and a GWAS meta-analysis of the components of multivariate longevity[18]. Using these data, we perform transcriptome-wide association studies (TWASs) and identify transcriptomic associations with EAA and multivariate longevity, including

novel (see "Methods" for definition) EAA-associated genes. We then employ fine-mapping and conditional analyses and prioritize high confidence (see "Methods" for definition) genes with potentially causal relationships with our aging-related traits. We functionally annotate these high confidence genes with Gene Ontology (GO) categories using Prediction of gene Insights from Stratified Mammalian gene co-EXPression (PrismEXP). Next, we perform cis-instrument Mendelian randomization (MR) of the druggable genome[19] and identify drug targets that could modify EAA and multivariate longevity. We also perform phenome-wide association studies (PheWASs) to identify potential pleiotropic effects of promising genetic drug targets. To identify circulating metabolites that impact aging-related traits, we conduct metabolome-wide MR analyses, prioritizing numerous metabolites that affect multivariate longevity. Finally, we perform cell-type enrichment analyses to identify cell types implicated in aging. The results of our cell-type enrichment analyses implicate immune cells in biological aging, and a follow-up MR analysis of 731 immune cell traits further elucidates the immune system's role in EAA and multivariate longevity. These findings may inform future research aimed at improving human aging.

## Results

### TWASs reveal transcriptomic architecture of aging traits

We used FUSION[20], cross-tissue expression quantitative trait locus (eQTL) weights[21], and GWAS summary statistics[10,18] to impute gene expression signatures associated with EAA and multivariate longevity. We report our full TWAS results in Supplementary Data 1–7, including results of colocalization analyses[22] and permutation testing[20] (Supplementary Data 1–5), conditional analyses[20] (Supplementary Data 6), and Fine-mapping Of CaUsal gene Sets (FOCUS)[23] (Supplementary Data 7). Our analyses identified 28 cross-tissue features significantly

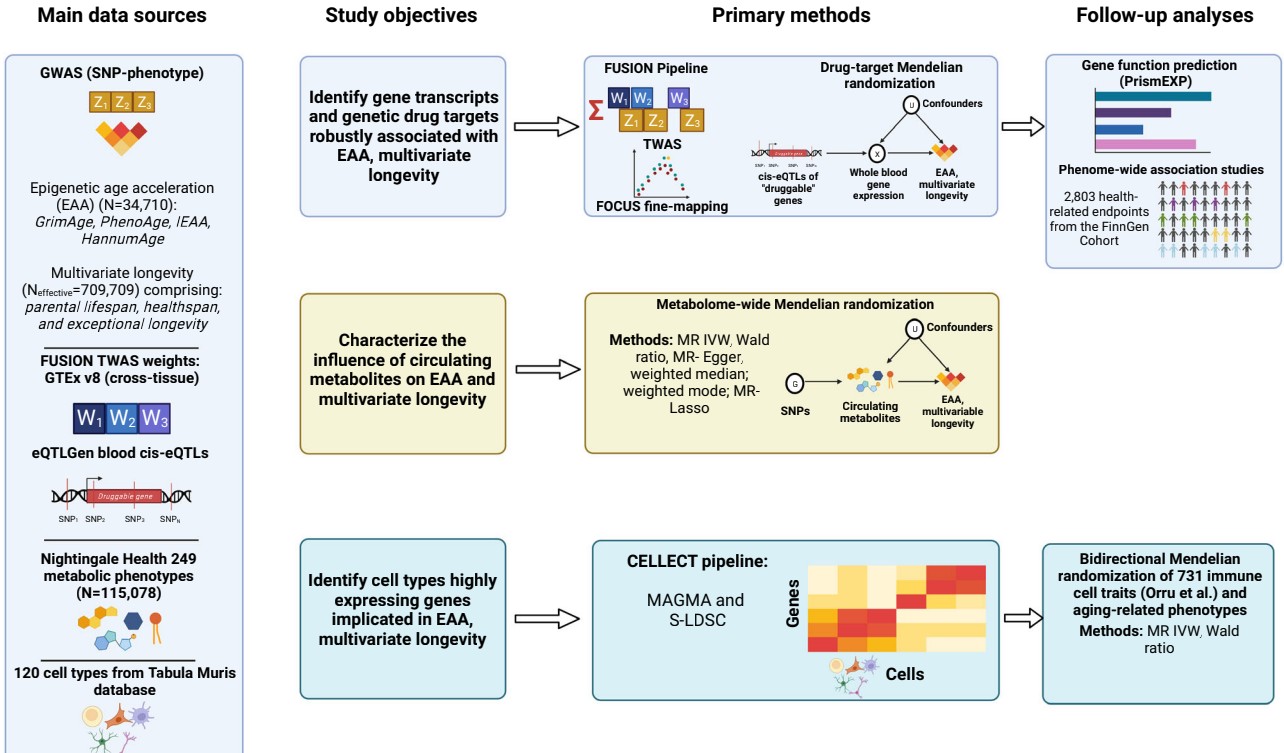

**Fig. 1 | Study overview.** An overview of this study's data sources, analytical flow, and methodology. Created with BioRender.com. IEAA intrinsic epigenetic age acceleration, TWAS transcriptome-wide association study, SNP single nucleotide polymorphism, eQTL expression quantitative trait loci, GTEx Genotype-Tissue Expression Project, IVW inverse variance weighted, CELLECT CELL-type Expression-specific

integration for Complex Traits, FOCUS Fine-mapping Of CaUsal gene Sets, MAGMA Multi-marker Analysis of GenoMic Annotation, S-LDSC stratified linkage disequilibrium score regression, PrismEXP Prediction of gene Insights from Stratified Mammalian gene co-EXPression.

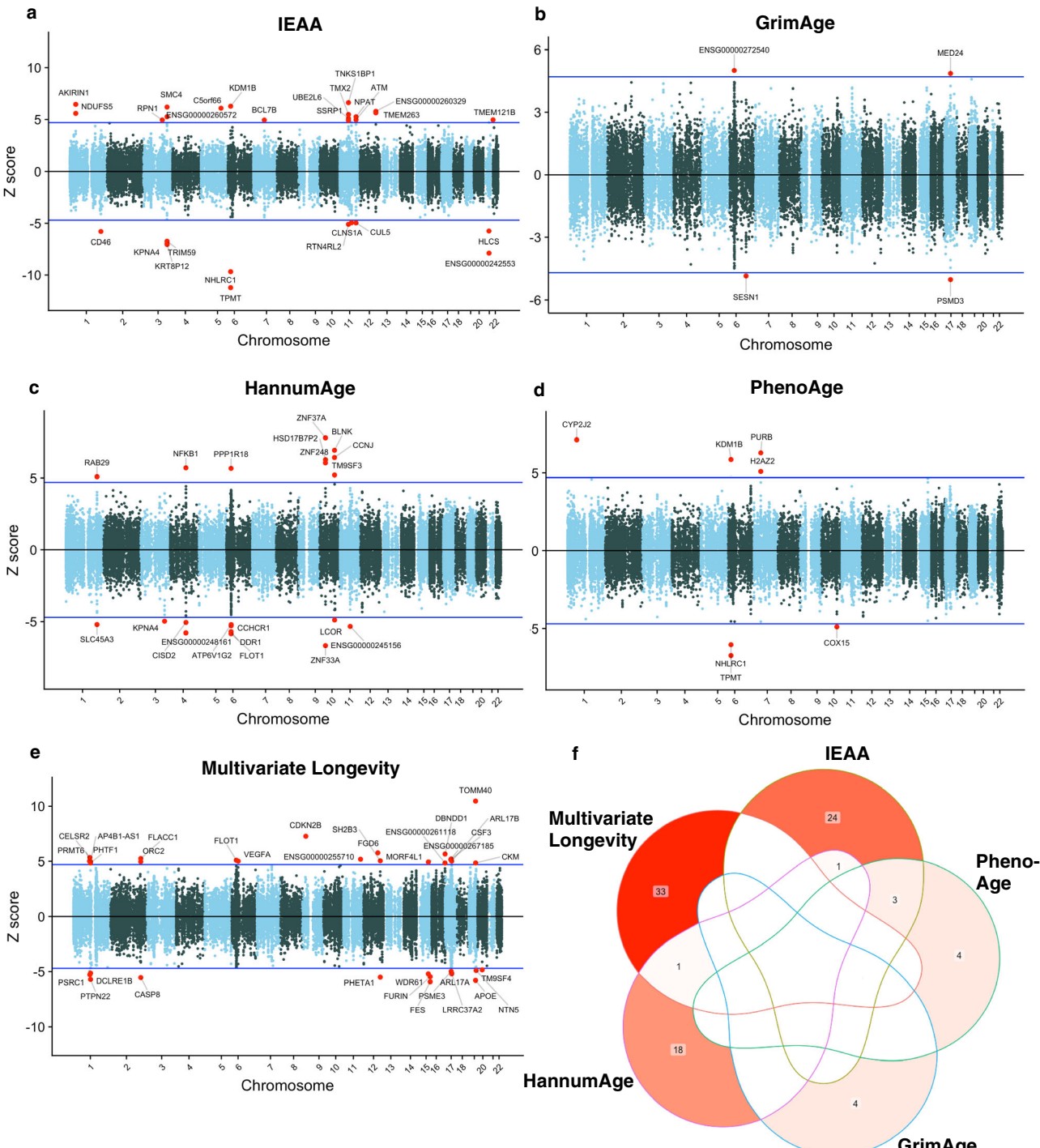

**Fig. 2 | Results of TWASs of EAA and multivariate longevity. a–e** Manhattan plots of gene-traits associations for aging-related traits (IEAA, GrimAge, HannumAge, PhenoAge, multivariate longevity). X axes represent genomic position. Blue lines represent $Z = 4.837$, which corresponds to a Bonferroni-corrected significance threshold of $P = 1.32 \times 10^{-6}$. Red circles represent statistically significant gene-trait associations. Statistical analyses were conducted using two-sided t-tests. **f** Venn diagram quantifying the overlapping genes shared by two or more aging-related phenotypes. Encircled numbers represent the number of significant genes shared between two or more phenotypes. TWAS transcriptome-wide association study, EAA epigenetic age acceleration, IEAA intrinsic epigenetic age acceleration.

associated with intrinsic epigenetic age acceleration (IEAA), 20 significantly associated with HannumAge, four significantly associated with GrimAge, seven significantly associated with PhenoAge, and 34 significantly associated with multivariate longevity after Bonferroni correction ($P < 1.32 \times 10^{-6}$) (Fig. 2). Most of these features colocalized with their respective aging phenotype, suggesting that a shared, pleiotropic SNP influences both gene expression and said aging phenotype (19/28 for IEAA, 8/20 for HannumAge, 4/4 for GrimAge, 4/7 for

PhenoAge, 21/34 for multivariate longevity). Moreover, of these significant features, the overwhelming majority passed conservative permutation testing, suggesting that these features represent bona fide signals rather than associations conditional on high GWAS signals. Additionally, 15 unique IEAA features, 13 unique HannumAge features, four unique GrimAge features, six unique PhenoAge features, and 20 unique multivariate longevity features passed conditional tests designed to identify, within a given locus, features independently

**Table 1 | High confidence genes associated with aging phenotypes (TWAS significant, conditionally significant, and PIP > 0.5)**

| Phenotype | Gene | Novel | TWAS Z score | FOCUS PIP | Joint P value (conditional analysis) | PP.H4 (colocalization analysis) | Permutation test P value |
|---|---|---|---|---|---|---|---|
| HannumAge | ZNF37A | NO | 7.80 | 1.00 | $5.30 \times 10^{-7}$ | 0.042 | 0.050 |
| | FLOT1 | YES | −5.83 | 0.67 | $3.30 \times 10^{-7}$ | 1.00 | 0.002 |
| | KPNA4 | YES | −4.97 | 0.83 | $6.90 \times 10^{-7}$ | 0.96 | 0.002 |
| | ZNF248 | NO | 6.05 | 1.00 | $2.40 \times 10^{-18}$ | 0.00 | 0.30 |
| | ENSG00000245156 | NO | −5.33 | 0.97 | $9.70 \times 10^{-8}$ | 0.88 | 0.008 |
| GrimAge | SESN1 | NO | −4.85 | 0.91 | $1.20 \times 10^{-6}$ | 0.94 | 0.022 |
| | ENSG00000272540 | NO | 5.00 | 0.72 | $5.60 \times 10^{-7}$ | 0.87 | 0.005 |
| IEAA | CD46 | NO | −5.80 | 1.00 | $6.60 \times 10^{-9}$ | 0.51 | 0.020 |
| | TPMT | NO | −11.50 | 1.00 | $9.20 \times 10^{-16}$ | 0.007 | 0.003 |
| | TNKS1BP1 | NO | 6.63 | 1.00 | $1.80 \times 10^{-7}$ | 0.80 | 0.003 |
| | RPN1 | NO | 4.95 | 0.92 | $7.30 \times 10^{-7}$ | 0.36 | 0.081 |
| | AKIRIN1 | NO | 6.47 | 0.98 | $9.90 \times 10^{-11}$ | 1.00 | $1.40 \times 10^{-4}$ |
| | TMEM121B | NO | 4.97 | 0.93 | $6.80 \times 10^{-7}$ | 1.00 | 0.002 |
| | NHLRC1 | NO | −9.68 | 1.00 | $3.70 \times 10^{-22}$ | 0.98 | 0.013 |
| | TMX2 | YES | 5.48 | 0.68 | $2.60 \times 10^{-4}$ | 0.30 | 0.026 |
| | KRT8P12 | NO | −7.04 | 0.51 | $1.90 \times 10^{-12}$ | 0.96 | $5.92 \times 10^{-4}$ |
| | ENSG00000260329 | NO | 5.79 | 1.00 | $7.10 \times 10^{-9}$ | 0.99 | 0.004 |
| PhenoAge | CYP2J2 | NO | 7.13 | 1.00 | $9.80 \times 10^{-13}$ | 0.97 | 0.002 |
| | TPMT | NO | −6.74 | 1.00 | $6.80 \times 10^{-7}$ | 0.007 | 0.002 |
| | PURB | NO | 6.29 | 1.00 | $3.10 \times 10^{-10}$ | 0.96 | 0.014 |
| | KDM1B | NO | 5.86 | 0.99 | $2.40 \times 10^{-4}$ | 0.01 | 0.019 |
| | NHLRC1 | NO | −6.03 | 1.00 | $1.60 \times 10^{-9}$ | 0.98 | 0.013 |
| Multivariate longevity | DBNDD1 | NO | 5.66 | 1.00 | $1.50 \times 10^{-8}$ | 1.00 | 0.009 |
| | TOMM40 | NO | 10.47 | 1.00 | $1.20 \times 10^{-25}$ | 0.20 | 0.004 |
| | CDKN2B | NO | 7.28 | 1.00 | $3.40 \times 10^{-13}$ | 0.88 | 0.002 |
| | FGD6 | NO | 5.77 | 1.00 | $8.20 \times 10^{-9}$ | 0.99 | $4.37 \times 10^{-4}$ |
| | FES | NO | −5.91 | 1.00 | $3.40 \times 10^{-9}$ | 0.98 | $4.51 \times 10^{-4}$ |
| | ENSG00000255710 | NO | 5.19 | 0.98 | $2.20 \times 10^{-7}$ | 0.95 | 0.019 |
| | PHETA1 | NO | −5.50 | 1.00 | $3.70 \times 10^{-8}$ | 0.078 | 0.009 |

High confidence results from TWAS analyses of five aging phenotypes. TWASs were conducted using cross-tissue expression weights generated from the GTEx v8 release using sparse canonical correlation analysis (sCCA). Significance was defined using a Bonferroni threshold of $P < 1.32 \times 10^{-6}$ (0.05/37,917 cross-tissue sCCA features). Significant TWAS associations were deemed high confidence if they passed a conditional test (joint P value < 0.05) and FOCUS fine-mapping (PIP > 0.5). Colocalization and permutation analyses were used to further assess the robustness of TWAS findings. A gene was defined as novel if it was located greater than 500 kilobases from a lead variant in the source GWAS. Statistical analyses were conducted using two-sided t-tests.
TWAS transcriptome-wide association study, PIP posterior inclusion probability, FOCUS Fine-mapping Of CaUsal gene Sets, PP.H4 posterior probability that two traits are associated with a single causal variant, IEAA intrinsic epigenetic age acceleration, GTEx v8 Genotype-Tissue Expression Project version 8, GWAS genome-wide association study.

associated with the trait of interest as opposed to feature-trait associations attributable to correlated expression between genes. Next, we used FOCUS fine-mapping to identify potentially causal, high confidence genes. We identified 10 high confidence features for IEAA, five for HannumAge, two for GrimAge, five for PhenoAge, and seven for multivariate longevity (Table 1). Two of these high confidence genes, both located on chromosome 6p22.3, appeared for multiple phenotypes: TPMT (IEAA, PhenoAge) and NHLRC1 (IEAA, PhenoAge). Additionally, two high confidence genes for HannumAge (FLOT1 and KPNA4) and one high confidence gene for IEAA (TMX2) were novel (>500 kilobases (kb) away from the nearest GWAS lead SNP). Thus, in total, 19/22 of our high confidence findings for EAA and 7/7 of our high confidence findings for multivariate longevity likely reflect signals from their respective source GWAS.

## High confidence TWAS genes are associated with diverse biological functions

We investigated the biological function of the high confidence genes identified by the TWAS pipeline using PRismEXP and the GO database[24]. We found our high confidence genes to be associated with a wide range of biological processes, cellular components, and

molecular functions (results presented in Supplementary Data 8–10). For example, SESN1 (a GrimAge high confidence gene) showed associations with several processes, including insulin response, nutrient sensing, and steroid metabolism. FLOT1 (a HannumAge high confidence gene) was associated with amyloid fibril formation and the cellular response to oxidative stress and reactive oxygen species. Except for IEAA and PhenoAge, which both have high confidence associations with TPMT and NHLRC1, we observed relatively limited overlap in the biological functions of genes associated with different aging traits (Supplementary Data 11). The few functions common to EAA and multivariate longevity genes included "negative regulation of gene expression" and "positive regulation of transcription, DNA-templated." Yet, while few specific GO gene sets were shared by genes associated with different aging traits, genes associated with each individual aging trait had functions broadly related to insulin signaling, mitochondrial function, cellular response to stress, and metabolism (Supplementary Data 11).

## MR identifies druggable genes that impact aging traits

We performed a drug-target MR analysis in parallel to our TWASs to inform anti-aging drug development (instruments presented in

**Table 2 | MR of the druggable genome identifies genes associated with aging phenotypes**

| Phenotype | MR method | Gene | Beta | SE | FDR-adjusted P value | PP.H4 (coloc SuSiE) |
|---|---|---|---|---|---|---|
| IEAA | Wald ratio | NHLRC1 | −1.84 | 0.40 | $4.76 \times 10^{-3}$ | 0.92 |
| HannumAge | Wald ratio | NFKB1 | −0.49 | 0.11 | 0.018 | 0.76 |
| | IVW | HDGF | −0.73 | 0.18 | 0.022 | 0.87 |
| PhenoAge | Weighted median | TPMT | 0.53 | 0.094 | $4.58 \times 10^{-5}$ | 1.00 |
| | IVW | LTBR | 0.46 | 0.11 | 0.031 | 0.85 |
| Multivariate longevity | Wald ratio | PSMA4 | −0.065 | 0.007 | $1.23 \times 10^{-19}$ | 0.93 |
| | Weighted median | CASP8 | 0.028 | 0.006 | $6.40 \times 10^{-4}$ | 1.00 |
| | IVW | VDR | 0.032 | 0.007 | 0.002 | 0.95 |
| | Weighted median | VDR | 0.032 | 0.008 | 0.012 | 0.95 |
| | Wald ratio | WNT3 | 0.038 | 0.010 | 0.020 | 0.90 |
| | Wald ratio | PTPN22 | −0.022 | 0.006 | 0.027 | 0.94 |
| | Wald ratio | CDC25A | 0.048 | 0.013 | 0.036 | 0.82 |
| | Wald ratio | CTSK | 0.013 | 0.004 | 0.036 | 1.00 |

Results from our drug-target MR analysis of the druggable genome[19] on five aging phenotypes. Genes significantly associated with an aging phenotype at a FDR of 0.05 and colocalized at PP.H4 > 0.75 are displayed. Standard errors are with respect to beta values. Statistical analyses were conducted using two-sided t-tests.
*MR* Mendelian randomization, *FDR* false discovery rate, *IVW* inverse variance weighted, *IEAA* intrinsic epigenetic age acceleration.

Supplementary Data 12, full results presented in Supplementary Data 13–17). For all druggable genes[19] with eQTLs within 10 kb, we used these eQTLs as instrumental variables to represent lifelong exposure to this gene's product and evaluated the effect of these exposures on our aging-related outcomes. We identified gene products significantly associated (false discovery rate (FDR) of 0.05) with four of our aging phenotypes (IEAA, HannumAge, PhenoAge, multivariate longevity). We identified four unique genetic drug targets for IEAA, six for HannumAge, two for PhenoAge, and 27 for multivariate longevity. Two genes were significantly associated with two phenotypes: *TPMT*, a high confidence TWAS gene for PhenoAge and IEAA, accelerated PhenoAge and IEAA; *C4B* accelerated HannumAge and decreased multivariate longevity. Also of note, *NHLRC1*, another high confidence TWAS gene for IEAA and PhenoAge that neighbors *TPMT*, significantly decelerated IEAA and decelerated PhenoAge at a FDR-adjusted P value of 0.084. *TPMT* encodes thiopurine S-methyltransferase, an enzyme that breaks down immunosuppressant thiopurine drugs; *NHLRC1* encodes malin, a ubiquitin ligase involved in the degradation of misfolded proteins and the regulation of glycogen; *C4B* encodes the basic form of complement factor 4, a part of the classical complement pathway (https://medlineplus.gov/genetics/gene/).

While our drug-target MR identified many significant associations, many of these associations, including the associations involving *C4B*, failed to show strong evidence of colocalization using the coloc Sum of Single Effects (SuSiE) regression framework (1/4 IEAA genes colocalized, 2/6 HannumAge genes, 2/2 PhenoAge genes, 7/27 multivariate longevity genes) (Supplementary Data 13–17). Non-colocalized gene-trait associations cannot be interpreted as causal relationships. Moreover, our gene most significantly associated with HannumAge, *CD248*, may be associated due to reverse causality, according to the MR Steiger test of directionality (Supplementary Data 14). Despite an overall lack of colocalization among our MR findings, *TPMT* and PhenoAge, as well as *NHLRC1* and IEAA, did show strong evidence of colocalization and thus may reflect causal relationships. Table 2 displays all colocalized drug-target MR findings.

**PheWASs elucidate pleiotropic effects of MR-identified genes**
To model potential effects of pharmacologically targeting our MR-identified genes, we used data from FinnGen[25] to carry out PheWASs. In Supplementary Data 18–22, we report associations significant under a Bonferroni-corrected threshold of $P < 1.78 \times 10^{-5}$. Notably, SNPs within 100 kb of our top multivariate longevity-associated gene, *PSMA4*, were

associated with decreased incidence of chronic obstructive pulmonary disease and lung cancer and increased the risk for coronary artery disease. SNPs located near the neighboring genes *TPMT* and *NHLRC1* were associated with decreased risk of atrial fibrillation, autoimmune and inflammatory diseases, and increased risk of sleep disorders. Broadly, many SNPs located near our MR-identified genes modulated endocrine, metabolic, and immune functions, incidence of respiratory disease, and incidence of musculoskeletal disease. Ultimately, follow-up studies should investigate the effects of modulating transcript levels of *TPMT*, *NHLRC1*, and other genes identified in this MR on biological aging, with particular attention to adverse and beneficial effects on the phenotypes identified in these PheWASs.

**MR identifies metabolomic effects on multivariate longevity**
To identify the effects of circulating metabolites on multivariate longevity and epigenetic aging, we performed a metabolome-wide MR analysis of 249 circulating metabolites[26] on our aging-related phenotypes (exposures described in Supplementary Data 23, and genetic instruments described in Supplementary Data 24). Inverse variance weighted (IVW) MR identified 160 metabolites with significant effects on multivariate longevity at a Bonferroni-corrected threshold of $P < 0.00122$ (Supplementary Data 25–29). The five most significant effects came from (1) ratio of apolipoprotein B (ApoB) to apolipoprotein A1 (ApoA1) ($\beta = -0.070$); (2) clinical low-density lipoprotein (LDL) cholesterol ($\beta = -0.071$); (3) phospholipids in small LDL ($\beta = -0.067$); (4) cholesteryl esters in medium very-low-density lipoprotein (VLDL) ($\beta = -0.068$); and (5) cholesterol in medium VLDL ($\beta = -0.066$) (Fig. 3). Many of these relationships were conserved across complementary MR methods (Supplementary Data 25–29). These findings seem to reflect the well-validated role of non-high-density lipoprotein (non-HDL) particles and associated apolipoproteins in the pathogenesis of cardiovascular diseases[27]. By contrast, we failed to identify any significant effects of circulating metabolites on EAA. The nominally significant associations we identified showed no indication that non-HDL lipoproteins and related metabolic phenotypes increase EAA.

**Cell-type enrichment analysis links cell types to aging traits**
We used CELL-type Expression-specific integration for Complex Traits (CELLECT)[28], with both Multi-marker Analysis of GenoMic Annotation (MAGMA)[29] and stratified linkage disequilibrium score regression (LDSC)[30] enrichment analysis tools, to identify etiological cell types for

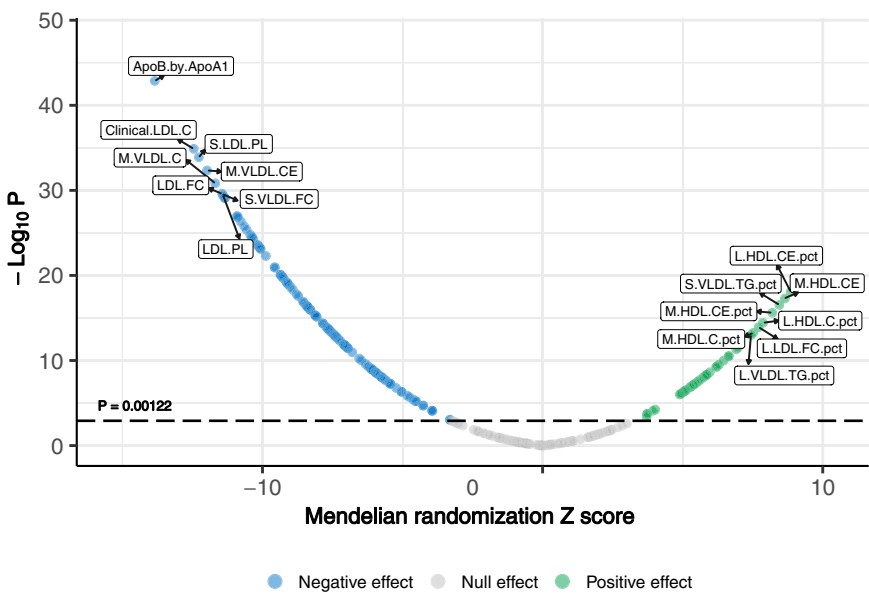

**Fig. 3 | Results of metabolome-wide MR analysis on multivariate longevity.** MR effects of metabolic phenotypes on multivariate longevity. Metabolic phenotypes with significant, positive Z scores (beta/standard error) are predicted to increase multivariate longevity and vice versa. The eight most significant positive and negative associations are labeled with abbreviated codes, and the full name corresponding to each code is contained in Supplementary Data 23. Green circles represent metabolic phenotypes that increase multivariate longevity and blue circles represent metabolic phenotypes that decrease multivariate longevity. The dotted line corresponds to a Bonferroni-adjusted significance threshold of $P = 0.00122$ (0.05/41 principal components). The full results of the metabolome-wide MR analysis, including estimates on EAA (all null), are contained in Supplementary Data 25–29. Statistical analyses were conducted using two-sided $t$-tests. MR Mendelian randomization, EAA epigenetic age acceleration.

multivariate longevity and EAA. Figure 4 displays the significant MAGMA findings, Supplementary Figs. 1–10 displays the results from all MAGMA and LDSC analyses, and Supplementary Data 30–34 contain full tabulated CELLECT results. The CELLECT results suggest that genes specifically expressed in immune cells are enriched in SNPs associated with EAA (Supplementary Figs. 1–4 and 6–9); most of the cell types significantly associated with EAA at a FDR of 0.05 are derived from marrow or the thymus, and even those derived from other tissues almost all play roles in innate or adaptive immunity. While no cell types were associated with multivariate longevity at a FDR of 0.05, CELLECT-MAGMA identified cell types associated with multivariate longevity at $P < 0.05$. These cells were also primarily immune cells (Supplementary Fig. 5). For the CELLECT-MAGMA analysis, the most significant cell-type enrichments overall were: (1) marrow common lymphoid progenitor cell ($P_{HannumAge} = 1.42 \times 10^{-8}$), (2) marrow late pro-B cell ($P_{HannumAge} = 1.20 \times 10^{-6}$), and (3) marrow late pro-B cell ($P_{GrimAge} = 2.26 \times 10^{-6}$). Interestingly, for both EAA and multivariate longevity, MAGMA produced a greater number of significant findings than LDSC. Additionally, across both methods, multivariate longevity had markedly fewer significant cell types than our EAA phenotypes, which may be partially due to the lower heritability of our multivariate longevity phenotype.

### MR identifies possible causal effects of immune traits on aging
To further elucidate the immune system's impact on multivariate longevity and epigenetic aging, we followed up our cell-type enrichments analyses with MR analyses of 731 immune cell traits[31] on multivariate longevity and EAA (traits contained in Supplementary Data 35, genetic instruments contained in Supplementary Data 36, and results contained in Supplementary Data 37–41). We identified 12 immune phenotypes with significant effects on multivariate longevity (FDR of 0.05): (1) Lymphocyte absolute count, (2) CD64 on CD14+CD16+ monocytes, (3) CD4+ T cell absolute count, (4) Central Memory CD4+ T cell count, (5) T cell absolute count, (6) HLA DR on CD33+ HLA DR+

CD14− myeloid cells, and (7) CD45 on B cells all negatively impact multivariate longevity. Conversely, (1) CD39 on CD39+ secreting CD4 regulatory T cells, (2) CD39 on CD39+ activated CD4 regulatory T cells, (3) CD14 on CD14 CD16- monocytes, (4) percentage of lymphocytes that are memory B cells, and (5) CD28 on CD39+ secreting CD4 regulatory T cells all positively impact multivariate longevity (Table 3, Supplementary Data 37). To identify immune traits that affect EAA, we used a relaxed FDR threshold of 0.2 because of the relatively lower sample sizes of our EAA samples. At this relaxed threshold, CD8 on terminally differentiated CD8+ T cells, CD80 on CD62L+ myeloid dendritic cells, and CD28 on CD28+ CD45RA+ CD8+ T cells were shown to increase IEAA (Table 3, Supplementary Data 38). None of these relationships were significant in the reverse direction at a FDR of 0.2 (i.e., MR of aging exposures on immune trait outcomes) (instruments contained in Supplementary Data 42, results contained in Supplementary Data 43–47). Overall, our MR results suggest that lymphocyte counts, cell surface molecule composition, and inflammatory processes impact multivariate longevity and may affect epigenetic aging.

## Discussion
Our study used large genomic data sources and a multi-omic approach to investigate biomolecular associations with EAA and multivariate longevity (Fig. 1). We identified novel gene associations with EAA and elucidated the transcriptomic effects of previously identified gene associations with EAA and multivariate longevity. Several highlighted genes (*TOMM40, SESN1, FLOT1, KPNA4,* and *TMX2*) were previously implicated in age-related endpoints including cancer[32–35], age-related clonal hematopoiesis[36], and cataract formation[36]. Furthermore, our study builds on past genetic studies that show overlap between aging-associated SNPs and SNPs involved in immune pathways[10,37]. We highlight the shared genetic control of aging and the immune system using CELLECT and MR, building a case for further research into the role of immunity and inflammation in the development of age-related diseases. Our study also identifies possible genetic drug targets,

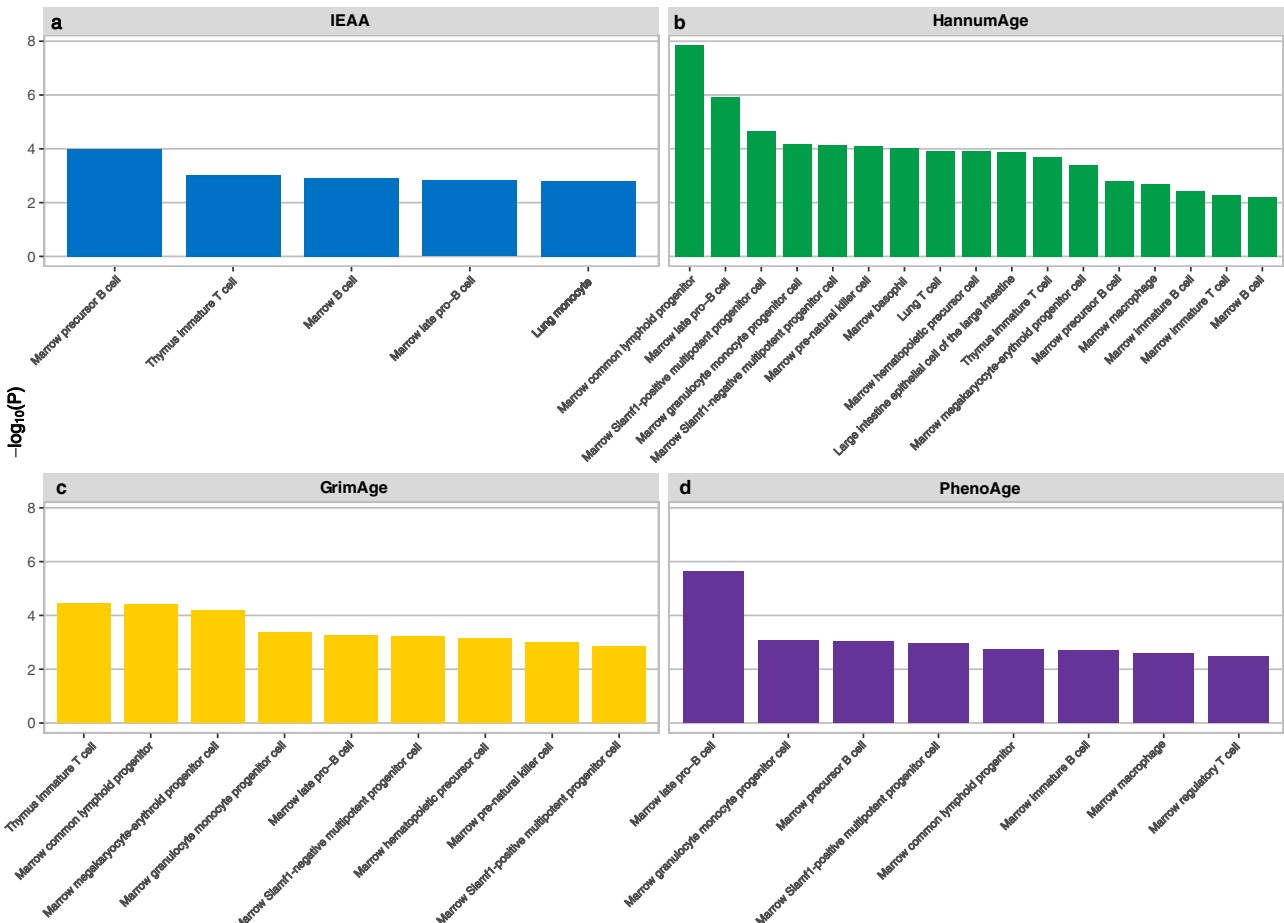

**Fig. 4 | CELLECT-MAGMA cellular associations with EAA. a–d** Results from CELLECT-MAGMA cell-type enrichment analysis of four EAA traits significant at a FDR of 0.05. Bars represent negative, log-transformed, unadjusted *P* values. The scRNA-seq data used in this analysis comes from the Tabula Muris database and encompasses 115 cell types from 20 *Mus musculus* tissues. The full results of the cell-type enrichment analyses, including CELLECT-LDSC results and cellular associations with multivariate longevity (all null), are contained in Supplementary Data 30–34. Statistical analyses were conducted using one-sided *t*-test tests. CEL-LECT CELL-type Expression-specific integration for Complex Traits; MAGMA Multi-marker Analysis of GenoMic Annotation, FDR false discovery rate, scRNA-seq single-cell RNA sequencing, LDSC linkage disequilibrium score regression.

including genes located at chromosome 6p22.3 (*TPMT* and *NHLRC1*), that warrant further study for their involvement in human longevity. Finally, we highlight several circulating metabolites that impact aging. Many of the aging-associated traits identified by this study, from genes to circulating metabolites, represent potential pharmacological targets. Because genomic evidence in drug discovery substantially improves the likelihood of successful drug development[38], these findings may facilitate basic and translational investigations into therapeutic strategies to increase healthy years lived.

Our TWASs identified many genes that have previously been linked with aging and age-related disease, supporting the validity of our approach. For instance, *SESN1*, a high confidence GrimAge gene, positively regulates lifespan in *Caenorhabditis elegans*[39] and inhibits the mammalian target of rapamycin protein (mTOR), a molecular process that has been widely investigated for its possible longevity benefits[40]. *TOMM40*, a high confidence multivariate longevity gene, was the most significant gene in a recent Alzheimer's disease TWAS[41] and influences age-related memory performance[42]. *APOE*, one of theour top multivariate longevity genes, has been associated with coronary artery disease, Alzheimer's disease, and longevity[43]. Our TWASs also revealed novel, high confidence associations between gene products and aging-related phenotypes. *FLOT1*, a novel, high confidence gene associated with HannumAge which encodes a marker of lipid rafts, is overexpressed in multiple cancers and negatively associates with cancer prognosis[32,33]. Downregulation of *FLOT1* has been shown to increase expression of *FOXO3*[44], a gene consistently implicated in human longevity[45]. *KPNA4*, a novel, high confidence HannumAge gene, has been implicated in cancer[34], age-related clonal haematopoiesis[36], and cataract formation[36]. *TMX2*, our final novel, high confidence gene, was associated with IEAA and is overexpressed in breast cancer[35].

Our cis-instrument MR analysis of druggable genes[19] identified two gene products that significantly modulate multiple age-related phenotypes. One of these genes, *C4B*, accelerated HannumAge and decreased multivariate longevity, but did not show strong evidence of colocalization with either phenotype. The other of these genes, *TPMT*, accelerated IEAA, albeit without evidence of colocalization, and accelerated PhenoAge with evidence of colocalization. Notably, TWAS also identified *TPMT* as a high confidence gene associated with IEAA and PhenoAge. *TPMT*'s product, thiopurine S-methyltransferase (TPMT), metabolizes thiopurine immunosuppressants[46]. Little data is available on the endogenous function of TPMT; its role in metabolizing immunosuppressants and our PheWAS findings linking SNPs near *TPMT* to immune-related disorders like post-dysenteric arthropathy and Guillain-Barre syndrome suggest it may enhance immune activity, while our GO analysis suggested that it may be involved in metabolism, mitochondrial function, and cellular response to oxidative stress.

**Table 3 | MR identifies immune cell traits associated with aging phenotypes**

| Phenotype | MR method | Exposure | Beta | SE | FDR-adjusted *P* value |
|---|---|---|---|---|---|
| Multivariate longevity | Wald ratio | Lymphocyte absolute count | −0.075 | 0.011 | $3.59 \times 10^{-9}$ |
| | Wald ratio | CD64 on CD14⁺CD16⁺ monocytes | −0.077 | 0.011 | $3.59 \times 10^{-9}$ |
| | Wald ratio | CD4+ T cell absolute count | −0.062 | 0.009 | $3.59 \times 10^{-9}$ |
| | Wald ratio | Central Memory CD4+ T cell count | −0.071 | 0.010 | $3.59 \times 10^{-9}$ |
| | Wald ratio | T cell absolute count | −0.078 | 0.011 | $3.59 \times 10^{-9}$ |
| | IVW | HLA DR on CD33+ HLA DR+ CD14− myeloid cells | −0.012 | 0.003 | $5.06 \times 10^{-4}$ |
| | IVW | CD45 on B cells | −0.033 | 0.009 | 0.038 |
| | IVW | CD39 on CD39+ secreting CD4 regulatory T cells | 0.005 | 0.001 | 0.038 |
| | IVW | CD39 on CD39+ activated CD4 regulatory T cells | 0.004 | 0.001 | 0.038 |
| | Wald ratio | CD14 on CD14 CD16- monocytes | 0.026 | 0.008 | 0.038 |
| | IVW | Percentage of lymphocytes that are memory B cells | 0.034 | 0.010 | 0.038 |
| | Wald ratio | CD28 on CD39+ secreting CD4 regulatory T cells | 0.013 | 0.004 | 0.048 |
| IEAA | IVW | CD8 on terminally differentiated CD8+ T cells | 0.33 | 0.085 | 0.053 |
| | IVW | CD80 on CD62L+ myeloid dendritic cells | 0.37 | 0.098 | 0.053 |
| | IVW | CD28 on CD28+ CD45RA+ CD8+ T cells | 0.34 | 0.094 | 0.053 |

Results from our MR analysis of the 731 immune trait exposures on five aging phenotypes. Genes significantly associated with multivariate longevity at a FDR of 0.05 or with EAA at a FDR of 0.20 are displayed. Standard errors are with respect to beta values. Statistical analyses were conducted using two-sided *t*-tests.
*MR* Mendelian randomization, *FDR* false discovery rate, *IVW* inverse variance weighted, *IEAA* intrinsic epigenetic age acceleration, *EAA* epigenetic age acceleration.

Although *TPMT* was prioritized by our transcriptomic analyses, it is important to note that colocalization results for these analyses yielded mixed evidence for a causal relationship between *TPMT* and EAA. *TPMT* resides in the 6p22.3 locus next to *NHLRC1*, a gene with high confidence TWAS associations with IEAA and PhenoAge and which significantly decelerated IEAA and nominally decelerated PhenoAge in our MR analyses. *NHLRC1* encodes malin, a ubiquitin ligase involved the regulation of glycogen[47]. *NHLRC1* showed strong evidence of colocalization with IEAA and PhenoAge in TWAS follow-up analyses that relied on the conservative single causal variant assumption[22], and it also showed evidence of colocalization with IEAA in our MR follow-up analysis. Because of the mixed evidence linking *TPMT* and *NHLRC1* to EAA outcomes, functional, fine-mapping, and interventional studies should investigate these genes and their role in aging.

Our metabolome-wide MR analysis identified 160 metabolic traits that significantly impact multivariate longevity, demonstrating an important role for the plasma metabolome in human longevity. Our most significant finding was a negative effect of the ratio of ApoB to ApoA1 on multivariate longevity, which extends MR data suggesting that elevated ApoB levels are associated with reduced lifespan[48]. The ApoB to ApoA1 ratio proxies the ratio of circulating non-HDL to HDL particles, and our findings thus support the notion that non-HDL particles are primary drivers of atherosclerosis[49]. Our results also highlight the important link between longevity and cardiovascular diseases, which remains the leading global cause of death[50]. However, while non-HDL particles may reduce longevity by promoting atherosclerosis, they did not accelerate epigenetic clocks in our analysis, suggesting recent observational data linking EAA with metabolomic dysregulation may be due to confounding[51].

Our CELLECT-MAGMA analyses revealed connections between immune cells and age-related traits, particularly EAA, demonstrating that aging-related genes are highly expressed in immune cells and their precursors, and thus, common genetic variants may impact EAA and multivariate longevity by influencing immunity and immune cell differentiation. Our CELLECT-LDSC results also weakly linked EAA to immune cells; however, this analysis did not suggest a relationship between immune cells and multivariate longevity. The discrepancies between MAGMA and LDSC may be due to the different ways by which the methods account for gene size and LD or other methodological

differences[29,30]. By highlighting general trends (e.g., immune enrichment) rather than specific findings in our cell-type enrichment analyses, we hope to mitigate the impact of methodological limitations.

Lastly, our downstream MR analysis of 731 immune-related exposures on multivariate longevity and EAA found that lymphocyte count, T cell count, and central memory CD4+ T cell count decrease multivariate longevity, while the proportion of lymphocytes that are memory B cells increase multivariate longevity. These findings align with emerging literature implicating CD4+ T lymphocytes as key players in an age-associated chronic inflammatory state linked to the pathogenesis of age-related diseases (inflammaging)[52,53]. We also identified immunological phenotypes related to lymphocyte cell surface molecule composition that impact EAA, particularly IEAA. Future studies should attempt to determine if and how the immune traits identified in this study influence inflammaging.

Our multi-omic analyses also facilitated biological comparisons between the five analyzed aging phenotypes. Firstly, this study revealed similarities and differences between the biological correlates of four prominent epigenetic clocks. Secondly, it juxtaposed EAA—a promising, yet recently developed and poorly understood biomarker for biological aging—and a multivariate longevity trait comprising phenotypes with clear relevance to human health and wellbeing.

At the transcriptomic level, there was one TWAS finding, *FLOT1*, shared between EAA (HannumAge) and multivariate longevity, which was only high confidence for HannumAge. Four TWAS associations were associated with multiple measures of EAA, including the high confidence associations of *TPMT* and *NHLRC1* with both IEAA and PhenoAge (Fig. 2). Functional analyses of our high confidence genes using GO revealed that, generally, genes implicated in each of our five aging phenotypes had different biological functions. The unique transcriptomic signatures of our four EAA phenotypes is particularly notable, yet not entirely unexpected, in the context of past research showing that different epigenetic clocks contain generally distinct CpG sites in distinct genomic regions[54]. We postulate that our four EAA measures may reflect unique epigenetic aging phenomena due to differences in training outcomes, tissues, and populations[9]. These epigenetic aging phenomena may capture various hallmarks of aging to different degrees. For example, PrismEXP implicated GrimAge (*SESN1*) and IEAA (*AKIRIN1*) genes in functions related to nutrient

sensing, while a HannumAge gene (*KPNA4*) was implicated in the regulation of gene expression. Dysregulation of each of these processes is considered to play a key role in age-related physiological decline[55,56]. However, we note that high confidence genes linked to each of our five aging outcomes were implicated in functions broadly related to insulin signaling, mitochondrial function, cellular response to stress, and metabolism. These biological domains may play fundamental roles in diverse aging-related phenotypes. Future in vivo and in vitro studies should attempt to better characterize the relationships of different epigenetic clocks and different biological process related to aging. Additionally, meta-clocks like the one described in Liu et al. (2020)[54], which may capture fundamental aging processes and predict aging-related health decline more effectively than individual component clocks, should be assessed as a potential aging biomarker.

Our metabolomic MR suggested that circulating lipids and lipoproteins strongly influence multivariate longevity but do not impact epigenetic aging (Fig. 3, Supplementary Data 25–29). These results indicate that any impact the circulating metabolome has on human longevity may not be mediated by facets of biological aging captured by epigenetic clocks.

By contrast, our cell-type enrichment analyses implicated immune cells and precursors in EAA to a greater extent than in multivariate longevity (Fig. 4, Supplementary Figs. 1–10). The relatively greater immune cell enrichment of EAA compared to multivariate longevity is particularly notable because our multivariate longevity dataset was larger than our EAA datasets. This result may be partially attributable to the greater heritability of our EAA phenotypes. In contrast to our cell-type enrichment analyses, our MR analyses of immune traits indicated that phenotypes related to lymphocytic makeup and central memory CD4+ T cell count may causally influence multivariate longevity with minimal effects on EAA. These findings could mean that a genetic predisposition for EAA exerts its effects on human longevity by altering immune function. However, describing relationships between EAA and immune-related traits is complicated by the fact that DNA methylation affects cellular differentiation and cell-type composition[57], making analyses of these traits potentially susceptible to reverse causality.

Our study has notable strengths. First, we enriched large GWAS datasets with functional significance through transcriptomic imputation. Our use of cross-tissue gene expression weights created using sparse canonical correlation analysis (sCCA) increased the statistical power of our TWASs[21], allowing us to detect more aging-related genes and capture the cross-tissue nature of aging. FUSION's post-processing tests helped us distinguish causal gene-trait associations from those resulting from a large GWAS signal or linkage disequilibrium (LD), and FOCUS fine-mapping allowed us to identify putative causal genes from among our TWAS findings. Our cis-instrument MR of the druggable genome served as a parallel transcriptomic technique, allowing us to identify potential longevity-promoting drug targets and prioritize consistent findings across transcriptomic methods. Finally, PheWASs allowed us to characterize potential side effects of drugs targeting MR-identified genes. Beyond transcriptomics, our study integrated GWAS data on EAA and multivariate longevity and single-cell transcriptomic data on diverse cell types to identify candidate etiological cell types for these aging phenotypes. Additionally, we took advantage of recent, comprehensive datasets to analyze the effects of 731 immune system traits and 249 circulating metabolites on multivariate longevity and EAA. Broadly, our study's hypothesis-free, comprehensive, multi-omic approach served to generate hypotheses for investigators seeking to understand and pharmacologically target the biology of aging. Our results provide insights into the biology of epigenetic clocks and present the biological signatures of these clocks in reference to a multivariate longevity phenotype encompassing healthspan, lifespan-by-proxy, and exceptional longevity, traits that any drug targeting biological aging processes would ultimately aim to modulate.

Our study also has important methodological limitations. First, our TWASs and MR analyses only used cis-eQTLs to predict gene expression, while trans-eQTLs and other elements also regulate gene expression. Our use of sCCA, while increasing our statistical power, likely obscured tissue-specific gene expression patterns that influence age-related phenotypes. For our cis-instrument MR of the druggable genome, we used eQTL data because of its more complete genome-wide and cross-tissue coverage compared to available protein QTL (pQTL) studies[58,59]. However, while transcriptomic data is more functional than genomic data, it is still one biological step removed from proteomic data, limiting our study's utility for drug development given that most approved pharmacotherapies target proteins[60]. Additionally, although we identified numerous high confidence genes, the techniques we used to account for LD, pleiotropy, and high GWAS signals are imperfect, and some of these genes may be false positives[61]. The importance of a cautious interpretation of even our high confidence findings is exemplified by the incongruous TWAS effect directions for genes like *FLOT1*, which is associated with decelerated HannumAge and greater multivariate longevity, but which nominally associates with accelerated IEAA and GrimAge. In summary, our findings should be used to generate hypotheses, and future in vitro, in vivo, and in silico studies should be used to validate and replicate them.

We also emphasize the limitations inherent to our data sources. For example, the GWAS of UK Biobank participants' parental lifespans may reflect the common causes of health and disease in the United Kingdom from several decades ago, which have shifted over time[62]. Also, due to a paucity of diverse -omic datasets, the human datasets we used throughout this study almost exclusively comprise participants of European ancestry. Due to differences in allele frequency, LD structure, and the genetic architecture of complex traits between ancestral populations, multi-omic analyses have limited trans-ancestral generalizability[63]. More comprehensive genomic data on health-related traits, gene expression, circulating metabolites, and immune phenotypes would allow studies like this one to be performed in non-European ancestry populations, avoiding the perpetuation of health disparities produced by ungeneralizable science[64]. Moreover, the GWASs of circulating metabolites and two of the three multivariate longevity phenotypes–healthspan and lifespan–included participants from the UK Biobank. MR analyses using these data therefore had sample overlap, which may have biased effect estimates toward their observational associations and away from the null[65], although recent literature suggests that two-sample MR methods can be safely used for one-sample MR in large biobanks[66]. Additionally, our Genotype-Tissue Expression Project version 8 (GTEx v8) cross-tissue gene expression weight dataset did not include expression weights for all genes that may be relevant to epigenetic aging or longevity. For instance, *TERT*, which was identified in GWASs of EAA[10,67] and the overexpression of which increases IEAA in primary human fibroblasts[67], was not contained within this reference data. Finally, due to the limited availability of broad human single-cell RNA-sequencing (scRNA-seq) datasets, we used *Mus musculus* scRNA-seq in our cell-type enrichment analysis, motivating our focus on cellular categories rather than specific cell types.

In conclusion, our study identified transcriptomic, metabolomic, and cellular correlates of EAA and a clinically relevant multivariate longevity phenotype; prioritized biological pathways relevant to aging; and identified drug targets that may reduce EAA and promote longevity. We also uncovered similarities and differences between epigenetic aging and multivariate longevity and between different epigenetic clocks. Ultimately, these findings may provide valuable insights into the transcriptomic, metabolic, cellular, immune-related architecture of age-related phenotypes and guide future research aimed at characterizing and pharmacologically combatting pathological aging.

## Methods

A study overview is presented in Fig. 1, including data sources, study objectives, methods, and follow-up analyses.

### Effect directions and sizes

Throughout this study, positive effects on multivariate longevity and negative effects on EAA should be interpreted as beneficial to human health. We use beta values to represent effect sizes across our MR analyses, which represent change in outcome phenotype per unit change in exposure phenotype. However, given the complexity of our aging phenotypes, we emphasize statistical significance and effect directions rather than effect sizes.

### Fundamental data sources

As the basis for all analyses in this study, we obtained publicly available summary statistics from GWASs of EAA and multivariate longevity conducted in European ancestry populations. These GWASs have existing ethical permissions from their respective institutional review boards, include participant informed consent and rigorous quality control, and are described fully in the following sections of "Methods". Because sex-stratified versions of these GWASs were unavailable, neither sex nor gender-based analyses were performed in this study. Additionally, to model LD in our MR, TWAS, and TWAS follow-up analyses, we used the 1000 Genomes Project Phase 3 European genomic reference data[68].

### EAA

To select SNPs associated with EAA for our multi-omic analyses, we used summary statistics from four GWASs of EAA performed by McCartney et al. in 28 European ancestry cohorts ($N = 34,710$)[10]. These epigenetic clocks use penalized linear regression models with DNA methylation data as an independent variable to predict chronological age (in the case of first-generation clocks including IEAA and HannumAge) or chronological age in addition to morbidity and mortality risk (in the case of second-generation clocks including GrimAge and PhenoAge)[10]. Specifically, HannumAge takes 71 CpGs as inputs and is trained on chronological age. IEAA takes 353 CPGs as inputs, is regressed on age and cell-type composition data imputed from this methylation data, and is trained on chronological age. PhenoAge takes 513 CpG sites as inputs and is trained on Phenotypic Age, which combines 9 biomarkers and chronological age to predict aging-related mortality. Finally, GrimAge incorporates 1030 CpG sites, age, sex, and methylation proxies for smoking, leptin, adrenomedullin, beta-2-microglobulin, cystatin C, growth differentiation factor 15, plasminogen activator inhibitor 1, and tissue inhibitor metalloproteinase 1.

### Multivariate longevity

To investigate a more clinically relevant aging phenotype and compare this phenotype to our EAA traits, we used data from a multivariate GWAS of traits related to human longevity ($N_{total} = 1,349,432$, $N_{effective} = 709,709$) for our multi-omic analyses. This recent GWAS[18] encapsulates healthspan ($N = 300,477$)[69], parental lifespan ($N = 1,012,240$)[70], and extreme longevity ($N = 36,745$)[71]. Briefly, the healthspan GWAS included 300,477 UK Biobank participants of European ancestry and used Cox-Gompertz survival models with clinical events in seven disease categories, including cancer, cardiovascular disease, diabetes, stroke, and dementia. Participants having one or more of these events were considered to have complete healthspans[69]. Of the 300,477 participants, 84,949 experienced an event which completed their healthspan[69]. Data from parental lifespan used information on 512,047 maternal and 500,196 paternal lifespans[69]. Across each cohort, Cox survival models for mothers and fathers were fitted, and the Martingale residuals of these survival models were regressed against participant gene dosages (imputed from the Haplotype Reference Consortium[72]). Finally, the extreme longevity GWAS used lifespan data from 11,262 unrelated participants of European ancestry who lived to an age greater than the 90th survival percentile compared to 25,483 participants whose age at death (or the last follow-up visit) was less than or equal to the 60th survival percentile[71].

### Gene expression weights for transcriptomic imputation

In our TWAS analyses, we sought to translate SNP associations with our aging outcomes into gene transcript associations with these outcomes in a tissue nonspecific manner to capture the cross-tissue nature of aging. Therefore, we used cross-tissue gene expression weights generated with sCCA[21]. These sCCA gene expression weights were derived from GTEx v8 atlas of eQTLs[73].

### Transcriptomic imputation

As the first step of our TWAS analyses, we used the munge_sumstats.py script from LDSC to appropriately format the GWAS summary statistics of our aging-related outcomes[74]. Next, we used the FUSION pipeline under default settings to impute transcriptomes associated with each of our outcomes[20]. This imputation was restricted to autosomal chromosomes. In brief, FUSION allowed us to (1) identify cis-heritable, cross-tissue gene expression features; (2) use our gene expression weights to develop SNP-based linear predictors of the expression level of each cis-heritable feature; and (3) calculate TWAS test-statistics based on these linear predictors and summary-level GWAS Z scores. FUSION selected the best gene expression model by comparing the out-of-sample $R^2$ value produced by several penalized linear regressions and Bayesian sparse linear mixed models (e.g., BLSMM, Elastic Net, LASSO, GBLUP)[20]. The statistical tests used in our TWASs and all other analyses in this study, with the exception of the cell-type enrichment analyses, were two-sided. To correct for multiple comparisons and allow for follow-up analysis on a manageable number of findings, we defined significance at a Bonferroni-corrected threshold of $P < 1.32 \times 10^{-6}$ (0.05/37,917 cross-tissue sCCA features). Finally, we considered our sCCA features to be novel if they were ≥500 kb away from the closest GWAS lead SNP.

### Conditional analyses, permutation testing, and colocalization

We performed follow-up analyses to assess the robustness of the gene transcript-trait associations we identified using TWAS. First, we used the FUSION suite's conditional tests to determine whether multiple TWAS-significant sCCA features within a given locus (±500 kb) were independently associated with our aging outcomes, or whether they were artifacts of a single feature-trait association produced by correlated expression between features[20]. To distinguish between these possibilities, the conditional analyses used the GWAS associations that remained after accounting for the predicted expression of other sCCA features in a given chromosomal locus to estimate the conditionally independent effect of each feature of interest. We defined conditional significance at a P value of 0.05. Additionally, to assess the possibility that the significance of our gene expression features was conditional on high GWAS effects, we carried out permutation testing for each locus[20]. We designed our test to stop after 100,000 permutations and defined significance at a P value of 0.05. Notably, the permutation testing statistic is highly conservative, and it is possible that truly causal genes can fail[20]. Finally, in line with previous TWAS studies[75,76], we performed colocalization of our TWAS-significant genes using the coloc R package (version 5.1.0.1)[22], implemented in FUSION. This allowed us to assess the probability that our TWAS associations reflected linkage between distinct causal SNPs (PP.H3) or a single causal SNP (PP.H4). Because traditional colocalization assumes that a single causal variant exists for a given trait in a specified genomic region[22], an assumption that is conservative and likely to be violated[77], we used it as a supplementary sensitivity analysis. We defined PP.H4 > 0.75 as strong evidence of colocalization.

## Fine-mapping of TWAS associations and high confidence findings

We employed FOCUS to identify genes that probably were responsible for the TWAS gene-trait association signal in a given locus, and thus likely had causal effects on the aging trait of interest[23]. FOCUS is a Bayesian model that accounts for correlation structures that result from LD, TWAS prediction weights, and pleiotropic SNP effects to identify gene sets containing a causal gene with 90% confidence (i.e., the credible set)[23]. Additionally, FOCUS determines posterior inclusion probabilities (PIPs) for individual features. A PIP > 0.5 indicates that the feature is the most likely causal feature within a risk region[23]. Importantly, unlike traditional colocalization, FOCUS performs well in scenarios in which, within a locus, multiple causal variants exist for a given gene, or multiple causal genes exist for a given trait[23]. In our study, we defined genes with a PIP > 0.5, a significant TWAS P value, and a significant conditional analysis P value as high confidence genes. High confidence genes are likely to have causal effects on EAA and/or multivariate longevity.

## Functional annotation of high confidence genes

We sought to characterize the biological function of the 27 unique high confidence genes that were associated with EAA or multivariate longevity by the transcriptomic imputation pipeline. Therefore, we performed a gene function prediction analysis[78] for each of the high confidence genes using the recently developed statistical method PrismEXP (Prediction of gene Insights from Stratified Mammalian gene co-EXPression), implemented in its Python package[79–81]. PrismEXP uses the ARCHS4 gene expression resource[81] to calculate predicted gene functions from gene set data[79–81]. We downloaded and analyzed the 2021 release of the GO[24] biological processes, molecular functions, and cellular components gene sets. Nine high confidence genes were not available in the GO gene sets (*ZNF37A, ZNF248, KRT8P12, CYP2J2, ENSG00000245156, ENSG00000272540, ENSG00000260329, ENSG00000255710*) and were therefore not included in the gene function prediction analyses. We defined significance for this analysis using Bonferroni-corrected P values based on the number of tested GO categories.

## eQTL data for drug-target MR

To generate genetic instruments for our drug-target MR analysis of aging phenotypes, we used a large eQTL dataset from whole blood ($N = 31,684$, European ancestry) from the eQTLGen Consortium[59]. This data included cis-eQTLs for 16,987 genes, defined by significance at a FDR of 0.05 across all 16,987 genes. We subset eQTLGen data based upon 4,479 genes comprising the druggable genome defined by Finan et al.[19]. The druggable genome comprises all genes encoding proteins targetable by currently available compounds, including compounds in clinical trials, approved medications, and small compounds validated in preclinical experiments[19]. We also investigated *NHLRC1*, a gene identified as high confidence by TWAS. We only included eQTLs located within 10 kb of the start or end base pair position of the gene of interest in our genetic instruments. This left 2714 genes to use as exposures in our drug-target MR analyses.

## Drug-target MR

All MR analyses in this study are reported according to the MR Strengthening the Reporting of Observational Studies in Epidemiology (STROBE) guidelines (Supplementary Note 1)[82]. We performed drug-target MR analyses to complement our TWAS findings and identify genes that could be targeted pharmacologically to increase multivariate longevity or decrease EAA. All MR analyses were performed using the R package TwoSampleMR v0.5.6[83]. First, we harmonized the eQTL instruments with each EAA and multivariate longevity GWAS and used the 1000 Genomes Project European reference sample to clump our instruments at LD $R^2 < 0.001$, ensuring independence between

SNPs included in genetic instruments[83]. We filtered out SNPs with F-statistics <10, the conventional threshold used to avoid weak instrument bias in MR studies, thus ensuring the use of strong instruments[84]. We also performed Steiger filtering to identify instrument SNPs that explain greater variation in an outcome than an exposure and performed MR with these potentially reverse causal SNPs both included and excluded. Such a SNP only appeared for two exposure-outcome pairs: *CD248* on HannumAge (discussed in "Results") and *KCNJ14* on GrimAge (insignificant regardless of this SNP's inclusion).

For eQTL instruments with 1 SNP, MR effect estimates were calculated using Wald ratios, while for instruments with 2 SNPs, we used the IVW MR estimator. For eQTL instruments with >2 SNPs, we the IVW MR estimator along with MR-Egger, weighted median, and weighted mode methods, which rely on weaker assumptions than IVW at the cost of lost statistical power[85]. These methods evaluate the sensitivity of MR findings to different patterns of violations of IVW assumptions, and concordant results using multiple methods strengthens causal inference[83]. We corrected for multiple comparisons by defining significance at a FDR of 0.05, defined by the number of unique gene-outcome combinations tested for each outcome. Findings remaining significant after this FDR correction were subject to colocalization analysis, performed with coloc using the SuSiE regression framework[86]. SuSiE relaxes traditional coloc's assumption of a single causal variant. We defined strong evidence for colocalization as a PP.H4 > 0.75 and reported posterior probabilities corresponding to the highest PP.H4 for each gene-trait association. We used traditional coloc as a supplementary sensitivity analysis[22]. We also performed the MR Steiger test of directionality for all exposure-outcome pairs to test for reverse causality[87]. Steiger P values below $10^{-250}$ are rounded to zero. Additionally, for each exposure-outcome pair, we performed the Egger intercept test to detect directional pleiotropic effects of each genetic instrument[88]. Finally, we performed the Cochran Q heterogeneity test to measure heterogeneity between variant-specific causal estimates for each instrument-outcome pair, as heterogeneity can indicate a violation of the MR assumption that instrument SNPs only operate through the exposure of interest[89].

## PheWASs

We evaluated genes that were significantly associated with an aging phenotype at a FDR of 0.05 in our MR analyses with PheWASs to identify possible health effects of pharmacologically targeting these genes. We leveraged publicly available electronic health record data from FinnGen Release 5[25], which features 218,792 patient records and 2803 health-related endpoints, to run our PheWASs. This patient database provided the advantage of being independent from other samples in our study. Using this data, we analyzed phenotypic associations of SNPs within 100 kb of each gene of interest. We reported all gene-trait associations significant under a Bonferroni-corrected threshold of $P < 1.78 \times 10^{-5}$ (0.05/2803).

## Metabolome-wide MR on aging traits

To identify metabolomic influences on healthy aging, we performed two-sample MR analyses of 249 circulating metabolic phenotypes on our four EAA measures and our multivariate longevity phenotype. We derived our metabolomic instrument data from the Nightingale Health-UK Biobank metabolomic GWAS partnership's first release ($N = 115,078$)[26]. Unless otherwise mentioned, our procedures for harmonization, clumping, instrument selection, and our MR methods and estimators were the same as those used for our drug-target MR. In this analysis, we selected SNP instruments at a conventional threshold of $P < 5 \times 10^{-8}$. Because the instruments for our metabolic phenotypes were well powered and the MR Steiger test of directionality showed strong evidence that our results reflected the correct causal direction (Supplementary Data 25–29), we did not employ Steiger filtering.

Because the Cochran Q heterogeneity test identified substantial heterogeneity in the effects of variants in our some of our metabolite instruments on multivariate longevity, we used MR-Lasso, implemented through the R package MendelianRandomization v0.6.0[90], to remove outlier variants and generate effect estimates less likely to be violating MR assumptions[91]. For cases in which we used MR-Lasso, we report the post-Lasso IVW effect estimate as the primary effect estimate. To define significance, we divided the conventional significance threshold of $P < 0.05$ by 41 principal components that explain 99% of variation in the levels of circulating metabolites, yielding a Bonferroni-corrected threshold of $P < 0.00122$ (https://github.com/nightingalehealth/ggforestplot/blob/master/vignettes/nmr-data-analysis-tutorial.Rmd).

### Cell-type enrichment analyses
To identify etiological cell types associated with our aging-related phenotypes, we integrated scRNA-seq data with the aforementioned GWAS summary statistics using CELLECT[28]. We derived our scRNA-seq data from Tabula Muris[92], a database containing transcriptomic data from 100,000 cells and 20 organs and tissues of *Mus musculus*. We downloaded and prepared this scRNA-seq data using CELLEX[28]. CELLEX calculates an expression specificity likelihood (ESμ) for each gene following normalization and pre-processing. We ran CELLECT on default settings using both MAGMA[29] and LDSC[30] enrichment analysis tools to identify cell types enriched in trait-associated genes. Specifically, MAGMA measures the extent to which genetic associations with a phenotype increase as a function of gene expression specificity for a given cell type. LDSC quantifies the extent to which SNP heritability for a given phenotype is enriched in the most specifically expressed genes for a given cell type. We manually categorized our cell types as immune cells/immune cell precursors or non-immune cells to evaluate relationships between the immune system and aging. To define significance, we used a FDR threshold of 0.05, calculated separately for each aging outcome.

### MR of immune cell phenotypes on aging traits
To further assess interactions between the immune system and aging-related phenotypes, we performed two-sample MR of 731 immune cell traits on four measures of epigenetic age acceleration and multivariate longevity. We used GWAS data from Orru et al. ($N = 3757$, European ancestry)[31] to generate our immune trait eQTL instruments. We used the same procedures for instrument selection, harmonization, clumping, and analysis that we used in our drug-target MR analyses. We used Wald ratios to calculate MR effect estimates when using genetic instruments with 1 SNP and the IVW method when using genetic instruments with 2+ SNPs. Because DNA methylation affects cell-type composition[57], we used these same MR methods to test the effects of our five aging phenotypes on the 731 immune cell phenotypes and assess the possibility of reverse causality. For both sets of MR analyses of the immune system, we defined significance using FDR thresholds based on 539 independent immune cell traits[31], calculated separately for each aging outcome.

### Reporting summary
Further information on research design is available in the Nature Portfolio Reporting Summary linked to this article.

## Data availability
All analyses in this study were conducted using publicly available data. URLs for the source datasets are as follows: epigenetic age acceleration GWAS summary statistics: https://datashare.ed.ac.uk/handle/10283/3645; multivariate longevity GWAS summary statistics: https://datashare.ed.ac.uk/handle/10283/3599; sCCA weights (used for transcriptomic imputation) and 1000 Genomes Project Phase 3 European genomic reference data (used for transcriptomic imputation and MR): http://gusevlab.org/projects/fusion/; eQTLgen whole blood eQTL data used for MR of the druggable genome: https://www.eqtlgen.org/; Nightingale metabolomics GWAS summary statistics used for MR: https://gwas.mrcieu.ac.uk/, batch: met-d; Immune cell trait GWAS summary statistics used for MR: https://gwas.mrcieu.ac.uk/, ebi-a-90001391 through ebi-a-90002121; scRNA-seq data used for cell-type enrichment analysis: https://tabula-muris.ds.czbiohub.org/; FinnGen R5 data used for PheWAS analyses: https://r5.finngen.fi/; 2021 GO biological processes, molecular functions, and cellular components gene sets used for PrismEXP analyses: https://maayanlab.cloud/Enrichr/#libraries. All data generated in this study upon which conclusions are based are available in the Supplementary Data. Source data are provided with this paper.

## Code availability
The software used in this study are available at the following online repositories. R package TwoSampleMR version 0.5.6[93]: https://mrcieu.github.io/TwoSampleMR/; R package MendelianRandomization version 0.6.0[94]: https://cran.r-project.org/web/packages/MendelianRandomization/index.html/; R package ggforestplot version 0.1.0 nmr-data-analysis-tutorial.Rmd: https://github.com/nightingalehealth/ggforestplot/blob/master/vignettes/nmr-data-analysis-tutorial.Rmd; Python package LDSC version 1.0.1 (https://github.com/bulik/ldsc); R FUSION pipeline, March 16th, 2020 version: http://gusevlab.org/projects/fusion/; Python package FOCUS version 0.6.10 (https://github.com/bogdanlab/focus); R package coloc version 5.1.0.1: https://cran.r-project.org/web/packages/coloc/index.html; Python package PrismEXP version 1.86[95]: https://github.com/MaayanLab/prismexp; Python package CELLECT version 1.3.0: https://github.com/perslab/CELLECT; Python package CELLEX version 1.2.1: https://github.com/perslab/CELLEX. Figure 1 was made using BioRender.com. Figure 2 was made using R package TWAS Plotter version 1.0: (https://github.com/opain/TWAS-plotter) and R package ggven version 0.1.8[96]: (https://github.com/yanlinlin82/ggvenn). Figure 3 and Supplementary Figs. 1–10 were made using R package EnhancedVolcano version 1.16.0: https://github.com/kevinblighe/EnhancedVolcano. Figure 4 was made using R package ggplot2 version 3.3.5: https://cloud.r-project.org/web/packages/ggplot2/index.html. R version 4.2.1 was used to format data for analyses.

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

## Acknowledgements

This work was supported by the National Institutes of Health (NIH) intramural funding [ZIA-AA000242 to F.W.L]; Division of Intramural Clinical and Biological Research of the National Institute on Alcohol Abuse and Alcoholism (NIAAA). Additionally, we want to acknowledge the participants and investigators of the studies used in this research without whom this effort would not be possible. We also acknowledge the Medical Research Council Integrative Epidemiology Unit (MRC-IEU, University of Bristol, UK), especially the developers of the MRC-IEU UK Biobank GWAS Pipeline. We acknowledge Nightingale Health and the UK Biobank for providing the metabolomic data used in this study. Finally, we acknowledge Ali M. Hamandi for assisting with proofreading.

## Author contributions

L.A.M., D.B.R., and F.W.L. designed the study. L.A.M. and D.B.R. performed the analyses. L.A.M., D.B.R., and F.W.L. interpreted the data. L.A.M. and D.B.R drafted the manuscript. F.W.L. supervised the study. L.A.M., D.B.R., A.S.B., J.J., J.W., and F.W.L. critically revised the study for important intellectual content and approved the final manuscript.

## Funding

## Competing interests

The authors declare no competing interests.
