## [Peer Review File · Nature Communications]

Multi-omic underpinnings of epigenetic aging and human longevityREVIEWER COMMENTS

Reviewer #1 (Remarks to the Author):

Mavromatis et al. performed a transcriptome-wide association scan (TWAS) and transcriptome-wide Mendelian randomisation (MR) analysis to identify genes and pathways underlying genetic variation in epigenetic ageing clocks and longevity phenotypes. They supplement these analyses with a cell-type enrichment analysis and MR analyses of blood metabolite measures and immune cell traits. The transcriptome element of their study reveals their ageing-related phenotypes are associated with expression of several genes (of which *TPMT* may be causal), while their downstream analyses highlight the immune system and non-HDL lipids as relevant ageing targets.

The immune system is a well-known ageing target that has previously been highlighted by a similar study which integrated gene expression data with a parental lifespan GWAS (PMID 32358504). Analogously, non-HDL lipids have repeatedly been implicated in ageing-related disease, including in a study using Mendelian randomisation of metabolite levels against parental lifespan (PMID 34704651). However, research into the genetic basis of epigenetic ageing clocks has been limited thus far, and Mavromatis et al. use existing datasets and well-established methods to provide a timely follow-up which expands previous research, presenting findings that may be of therapeutic importance.

I do have some concerns about invalid gene set enrichment analyses (GSEA), confounding by linkage disequilibrium (LD), and confounding by reverse causality, each of which may reduce the reliability of their findings.

Main concerns:

1. The selection of input genes for GSEA and the type of enrichment used may not be valid. The FUMA software uses a hypergeometric test to assess whether input genes are overrepresented in predefined gene sets. This method assumes input genes are independent and weights them all equally. Applying FUMA to TWAS results has two issues:

1.1. TWAS is known to highlight clusters of genes which are correlated due to LD (PMID 30926968). Specifically, genetically predicted gene expression levels can be correlated when the genetic predictors (i.e. *cis*-eQTL) are shared or in high LD between genes (resulting in genetic correlation even when phenotypic expression levels are not correlated). Using correlated genes as input in a hypergeometric test will inflate gene set enrichment statistics for any set containing the correlated genes. This inflation may be especially prominent in positional gene sets, which are more likely to contain genes correlated due to LD.

1.2. The authors use all genes meeting a TWAS P value threshold of $5E-4$ as input for FUMA. However, this cutoff is arbitrary and results in a loss of information: TWAS P values or Z scores could be used to provide additional granularity in the enrichment analysis. A rank-based gene set enrichment analysis may therefore be more appropriate.

To circumvent both of the aforementioned issues, the authors should use a GSEA method that can use a ranked list of correlated genes, such as TWAS-GSEA (PMID 31230729).

2. It is conventional to perform a colocalisation analysis of TWAS-significant genes to establish whether genes are confounded by horizontal pleiotropy due to LD linkage (e.g. PMID 31230729, 33279206, 34704651). The authors should use coloc or similar software to see whether the genome-wide SNP associations for expression of the gene of interest and the ageing-related outcome share one (or more) causal variant(s). The same principle applies to MR analyses, where each *cis*-eQTL used to instrument gene expression levels should be confirmed to colocalise with the outcome in order to be valid instrument (see e.g. PMID 35452592)

3. Associations between *cis*-eQTL and traits have been shown to suffer from reverse causality (PMID 34561431), which can confound MR effect size estimates. Analogously, DNA methylation

patterns are dependent on sampled cell types (PMID 22568884), and therefore, MR analysis of immune cell types on epigenetic clocks could be confounded through reverse causality. Importantly, the authors' assertion that "epigenetic aging... may be more intertwined with immune function than measures like healthspan and lifespan" is far too strong given the potential for reverse causality influencing the epigenetic ageing phenotypes. The authors should acknowledge this limitation, and ideally, test to what extent reverse causality plays a role in their MR findings using bidirectional MR (or similar methods).

Minor comments:

4. The word "longevity" in the context of GWAS is understood to refer to an exceptional survival phenotype (PMID 25814633). The authors use of "longevity GWAS", or more misleadingly, "GWAS meta-analysis of longevity", can easily be misinterpreted as a (meta-analysis of multiple) GWAS of exceptional survival. The GWAS in question is actually a multivariate analysis of three distinct longevity-related phenotypes (parental lifespan, healthspan, and longevity). I encourage the authors to consider using more nuanced language when referring to this GWAS throughout their manuscript.

5. The sample size of the multivariate longevity GWAS is cited to be $N = 1,249,465$, but according to the original article, it should be either $N = 1,349,432$ for total sample size or $N = 709,709$ for the effective sample size (PMID 32678081). Regardless, given this dataset is composed of GWAS with different study designs, it would be more transparent to list the sample size of each study separately rather than summarising the dataset with a single number.

6. The authors present a list of TWAS genes they deem "high-confidence", acknowledging only a few of these genes are significant in multiple ageing-related GWAS. However, looking at TWAS statistics, it appears some of the genes do not even have concordant signs of effect between phenotypes. For example, TWAS Z scores for the newly highlighted FLOT1 gene are HannumAge -5.83 , Longevity 5.09 , GrimAge 4.44 , IEAA 2.39 , PhenoAge -2.16 , (i.e. pro-longevity in HannumAge/PhenoAge/Longevity, and anti-longevity in IEAA/GrimAge). The authors should take these conflicting statistics into account when discussing the reliability of their results.

7. While the authors use a suite of standard MR models when performing *cis*-eQTL MR, they only report effect sizes and P values regarding the MR slope. However, it would be useful to see statistics regarding the MR-Egger intercepts in a table, as a significant MR-Egger intercept can indicate the presence of directional horizontal pleiotropy. For the metabolome-wide and cell-type MR analyses, the authors should report MR-Egger intercept standard errors and P values in addition to the MR-Egger intercept estimates which are currently listed in their tables.

8. Why were MR Steiger tests performed only for exposure-outcome pairs with more than one genetic instrument? It is my understanding the test can be performed with a single SNP as well.

9. Strictly speaking, the MR analysis of metabolite measures on the multivariate longevity GWAS is not a two-sample analysis: UK Biobank samples were used both in selecting the exposure instruments and in calculating two of the three outcome GWAS statistics. The authors should acknowledge sample overlap will bias MR effect sizes towards the observational correlation between exposures and longevity, rather than the null (increasing false positive findings). In addition, the authors adjust for multiple testing of metabolites using FDR correction, yet this correction assumes each hypothesis test is independent. As some metabolite traits are ratios of other traits, this independence assumption clearly does not hold. A more appropriate method to correct for multiple testing would be to estimate the number of independent metabolite traits (using e.g. principal component analysis) and adjust for this estimated number instead.

10. The authors have a section on computational drug repurposing and drug set enrichment, but only list its methods and results in a supplementary file without any mention in the main text. As these analyses are not central to the current study and suffer from the same enrichment bias due to

correlated TWAS statistics (see concern 1.1 above), I politely question why this section was included in the manuscript. The authors should either list and discuss (a summary of) these supplementary results in the main text, or remove them from the study.

11. Please specify whether the heritability estimates cited in the introduction are SNP-based or pedigree-based, and double-check the numbers cited are correct. From my understanding, the lower bound of SNP-based heritability should be 0.10, as estimated for GrimAge acceleration in McCartney et al. (PMID 34187551; Table S19), whereas the upper bound of pedigree-based heritability estimates should be 0.43 (PMID 25633388).

12. In Supplementary Tables 1-5, any TWAS results with $P > 0.05$ have permutation P values of zero. If these TWAS results were not tested using the permutation method, these statistics should be set to NA rather than zero to avoid confusion.

13. Figure 2 and Figure 3, which contain the results from metabolome-wide MR and cell-type enrichment respectively, do not appear to be very informative. I would prefer the authors focus on a summary of the significant MR effects, rather than displaying the full results for all longevity traits and metabolites/cell-types. For example, in the metabolome MR, the effects of cholesterol-to-lipids ratio traits are largely captured by cholesterol and lipid traits, and the only significant MR effects for these categories are on the multivariate longevity phenotype. The authors may want to consider an alternative way to highlight the most pertinent findings from these analyses.

14. Reference 6 and 48 refer to the preprint and published versions of the same Gibson et al. paper. Ditto for reference 9 and 15 for the McCartney et al. paper. The published versions of these studies should be cited, unless there is a good reason to also cite the preprints.

Reviewer #2 (Remarks to the Author):

Mavromatis and colleagues conducted a comprehensive multi-omic analysis to address the underlying mechanisms of aging and to identify druggable molecular targets (TPMT is highly relevant). Leveraging large publicly available datasets and computational resources, the study integrated genome, transcriptome, and metabolomics and identified new genes for epigenetic age acceleration and longevity. The results highlight the important roles of immune and inflammation in the aging process. The study is timely and the findings, if replicated, may add new knowledge for aging-related diseases.

Comments

1. Identifying highly confident genes using FOCUS is a strength of the study. Some high confidence genes by TWAS are relevant to aging based on previous research. However, it is puzzling that no gene is shared among the four EAA phenotypes measured by epigenetic clocks, and only one gene overlapped with longevity. This raises the question of whether the four clocks measured the same trait. I would appreciate elaboration or further discussion of this question.

2. Although the positional gene set of chr6p21 is common among the three traits, less than 50% of the significant pathways are shared across different measures of aging and longevity. Thus, these discrepancies should be discussed. The region of chr6p21 has been linked to many pathological conditions so it is not surprise to see the enrichment with mortality and longevity.

3. The imputation of the transcriptome is built on the previous large GWAS on EAA by McCartney et al., which reported 137 significant loci. What proportion of the loci is shared between TWAS in this analysis and the GWAS on EAA? The significance of the common genes or eQTLs should be examined.

4. In Figure 3, many metabolites from the MR results for epigenetic age and longevity illustrate different directions. Please explain this discrepancy.

5. A previous study reported that TERT is associated with IEAA and EEAA. However, this gene is not replicated in this large study.

6. Importantly, besides European ancestry samples, 6,482 African American individuals were included in the McCarthney's paper. It would be helpful to report the TWAS findings and other similar analyses in AA ancestry samples so the results may be more generalizable, especially for identifying the druggable genes.

Reviewer #3 (Remarks to the Author):

I really struggled to follow this paper. Maybe I am not the intended audience for this manuscript, but I didn't take much from this work - for me, it was a disjointed set of analyses with no obvious prior hypothesis, and no strong conclusion. I appreciate that this hypothesis-free discovery approach is one way to do science, and maybe there are others for whom these findings are interesting - maybe for some, this is a goldmine or treasure trove of information. So please feel free to ignore this review if those who are part of the intended audience appreciate it, but honestly - I doubt if I'll remember anything from reading this paper even in a week's time.

Some comments:

1. I appreciate that in places the authors have tried hard to make their work understandable, but in other places, the approach taken and the motivation for this was unclear. As an example, the first substantive sentence of the methods reads: "Our TWAS analyses used pre-computed, cross-tissue, single nucleotide polymorphism (SNP) expression quantitative trait loci (eQTL) weights generated using sparse canonical correlation analysis (sCCA)." While I understand most of these words individually, their combination is beyond me. An explanatory phrase to motivate would help, eg: "To summarize the data at each gene region into a single variable, our TWAS analyses...". Or whatever the motivation for this is. Please remember - the point of a scientific paper is not to dazzle the reader with your ability to form complex sentences, but to communicate, and to be understood.

2. What do the points represent in Figure 3? I don't understand this figure - why is there more than one point per metabolite/outcome pair?

3. Did you consider colocalization for the top drug-target hits?

4. Everything is done in such a high-throughput way that it's difficult to focus on any one analysis, but I was slightly concerned about the MR methods that exclude variants from the analysis (MR-Lasso and penalized weighted median). If this method is excluding 1 or 2 variants out of 10-20, then that's fair enough. But if it is excluding a substantial fraction of the variants from the analysis, then it may give a false impression of consistency in the analysis - everything is consistent if you remove enough of the heterogeneity. I appreciate this isn't practical in each case, but it'd be nice to know what % of variants are being excluded from the analysis for the key metabolites.

5. My final comment is a somewhat broad and probably unhelpful comment about ageing as an outcome. If you are developing a drug to lower blood pressure, then everything is simple - you take the drug, and measure blood pressure in a week - did it go down or not? Does the drug work? - you should get a definitive answer in quick order. If you are developing a drug to slow the ageing process, then validating the drug is much harder. And if you can't validate the drug, then what is the point in investigating the mechanisms? I realise this is an unfair question, as scientific research shouldn't only be undertaken because it is practically useful. But I did wonder if you could comment on the relative value of an investigation into longevity as opposed to an investigation into the common components of longevity (eg cardiovascular disease, cancer, dementia), which are more likely to have a direct biological mechanism, as well as a potential target population for any drug developed.

January 13th, 2023

Dear Reviewers,

Thank you for taking the time to review our manuscript, “Multi-omic underpinnings of epigenetic aging and human longevity.” We have amended the title of our manuscript to be more concise and fulfill editorial guidelines. Each of you provided valuable feedback. Thanks to this feedback, our revised manuscript is more focused on key findings, better communicated through language and figures, and more analytically comprehensive than our initial submission. Please find our point-by-point responses to your comments below.

Sincerely,

Falk W. Lohoff, M.D.

Point-by-point response:

Reviewer #1 (Remarks to the Author):

Mavromatis et al. performed a transcriptome-wide association scan (TWAS) and transcriptome-wide Mendelian randomisation (MR) analysis to identify genes and pathways underlying genetic variation in epigenetic ageing clocks and longevity phenotypes. They supplement these analyses with a cell-type enrichment analysis and MR analyses of blood metabolite measures and immune cell traits. The transcriptome element of their study reveals their ageing-related phenotypes are associated with expression of several genes (of which *TPMT* may be causal), while their downstream analyses highlight the immune system and non-HDL lipids as relevant ageing targets.

The immune system is a well-known ageing target that has previously been highlighted by a similar study which integrated gene expression data with a parental lifespan GWAS (PMID 32358504). Analogously, non-HDL lipids have repeatedly been implicated in ageing-related disease, including in a study using Mendelian randomisation of metabolite levels against parental lifespan (PMID 34704651). However, research into the genetic basis of epigenetic ageing clocks has been limited thus far, and Mavromatis et al. use existing datasets and well-established methods to provide a timely follow-up which expands previous research, presenting findings that may be of therapeutic importance.

I do have some concerns about invalid gene set enrichment analyses (GSEA), confounding by linkage disequilibrium (LD), and confounding by reverse causality, each of which may reduce the reliability of their findings.

Main concerns:

1. The selection of input genes for GSEA and the type of enrichment used may not be valid. The FUMA software uses a hypergeometric test to assess whether input genes are

overrepresented in predefined gene sets. This method assumes input genes are independent and weights them all equally. Applying FUMA to TWAS results has two issues:

1.1. TWAS is known to highlight clusters of genes which are correlated due to LD (PMID 30926968). Specifically, genetically predicted gene expression levels can be correlated when the genetic predictors (i.e. *cis*-eQTL) are shared or in high LD between genes (resulting in genetic correlation even when phenotypic expression levels are not correlated). Using correlated genes as input in a hypergeometric test will inflate gene set enrichment statistics for any set containing the correlated genes. This inflation may be especially prominent in positional gene sets, which are more likely to contain genes correlated due to LD.

1.2. The authors use all genes meeting a TWAS P value threshold of 5E-4 as input for FUMA. However, this cutoff is arbitrary and results in a loss of information: TWAS P values or Z scores could be used to provide additional granularity in the enrichment analysis. A rank-based gene set enrichment analysis may therefore be more appropriate. To circumvent both of the aforementioned issues, the authors should use a GSEA method that can use a ranked list of correlated genes, such as TWAS-GSEA (PMID 31230729).

Author response: Thank you for the suggestion and for highlighting these important aspects of TWAS (gene clustering and gene correlation in GSEA).

We performed several complementary analyses as part of the FUSION pipeline (conditional testing, fine mapping, etc.) to identify a list of “high confidence” TWAS genes associated with epigenetic age acceleration and multivariate longevity. Therefore, to better characterize the biological function of these high confidence genes and circumvent the methodological issues that you mentioned, we have replaced the GSEA with a downstream gene function analysis of only these high confidence genes. For this analysis, we use PrismEXP (<https://github.com/MaayanLab/prismexp>) in conjunction with data from Gene Ontology’s biological processes, molecular functions, and cellular components gene sets to functionally annotate our 28 unique high confidence genes. Although we believe that our high confidence genes are robustly associated with our aging traits, to further protect against inflated enrichment statistics, we opted not to query positional gene sets in our PrismEXP analysis. We describe our PrismEXP analysis in the relevant section of the Methods, describe its results in the relevant sections of the Results, and discuss its implications in paragraph 8 of the Discussion.

We hope that this change enhances the focus of our manuscript while eliminating the methodological issues posed to GSEA by gene clustering and the use of a relaxed P value threshold.

2. It is conventional to perform a colocalisation analysis of TWAS-significant genes to establish whether genes are confounded by horizontal pleiotropy due to LD linkage (e.g. PMID 31230729, 33279206, 34704651). The authors should use coloc or similar software to see whether the genome-wide SNP associations for expression of the gene of interest and the ageing-related outcome share one (or more) causal variant(s). The same principle applies to MR analyses, where each *cis*-eQTL used to instrument gene expression levels should be confirmed to colocalise with the outcome in order to be valid instrument (see e.g. PMID 35452592).

Author response: Thank you for the suggestion. We performed colocalization analyses of our significant MR and TWAS findings. The results are presented in full in Supplementary Data 1-5 (TWAS) and Supplementary Data 12-16 (MR) and described as follows in the Methods section of the manuscript:

Post-TWAS colocalization: *“Finally, in line with previous TWAS studies,^{76,77} we performed colocalization of our TWAS-significant genes using the coloc R package (version 5.1.0.1),²⁰ implemented in FUSION. This allowed us to assess the probability that our TWAS associations reflected linkage between distinct causal SNPs (PP.H3) or a single causal SNP (PP.H4). We defined $PP.H4 > 0.75$ as strong evidence of colocalization.”*

Post-MR colocalization: *“Findings remaining significant after this FDR correction were subject to colocalization analysis, which used the same methods²⁰ and interpretation as our post-TWAS colocalization analysis.”*

Most of our TWAS findings showed strong evidence of colocalization (19/28 IEAA-associated genes, 8/20 for HannumAge, 4/4 for GrimAge, 4/7 for PhenoAge, 21/34 for multivariate). By contrast, many of our MR findings failed to show strong evidence of colocalization (0/4 IEAA-associated genes, 1/6 for HannumAge, 1/2 for PhenoAge, 6/27 for multivariate longevity). We describe these findings in the Results as follows:

Post-TWAS colocalization: *“Most of these features colocalized with their respective aging phenotype, suggesting that one or more shared, pleiotropic SNP influences both gene expression and said aging phenotype (19/28 for IEAA, 8/20 for HannumAge, 4/4 for GrimAge, 4/7 for PhenoAge, 21/34 for multivariate longevity).”*

Post-MR colocalization: *“However, few of the genes identified by the drug-target MR analysis, including neither TPMT, C4B, nor NHLRC1, demonstrated strong evidence of colocalization with our aging phenotypes (0/4 IEAA genes colocalized, 1/6 HannumAge genes, 1/2 PhenoAge genes, 6/27 multivariate longevity genes) (Supplementary Data 13-17).”*

We discuss the implications of our colocalization findings with respect to two of our high confidence gene findings, *TPMT* and *NHLRC1*, in the Discussion:

“Although TPMT was prioritized by our transcriptomic analyses, it is important to note that colocalization results for our TWAS and drug-target MR analyses of TPMT yielded no evidence for a causal relationship between TPMT and EAA. TPMT resides in the 6p22.3 locus next to NHLRC1, a gene with high-confidence TWAS associations with IEAA and PhenoAge and which significantly decelerated IEAA and nominally decelerated PhenoAge, according to our MR results. NHLRC1 encodes malin, a ubiquitin ligase involved in the regulation of glycogen.⁴⁷ NHLRC1 showed strong evidence of colocalization with IEAA and PhenoAge in TWAS follow-up analyses but failed to show evidence of colocalization with these traits in MR follow-up analyses. Because of the mixed evidence linking TPMT and NHLRC1 to EAA outcomes, functional, fine-mapping, and interventional studies should investigate these genes and their role in aging.”

3. Associations between *cis*-eQTL and traits have been shown to suffer from reverse causality (PMID 34561431), which can confound MR effect size estimates. Analogously, DNA methylation patterns are dependent on sampled cell types (PMID 22568884), and therefore, MR analysis of immune cell types on epigenetic clocks could be confounded through reverse causality. Importantly, the authors' assertion that "epigenetic aging... may be more intertwined with immune function than measures like healthspan and lifespan" is far too strong given the potential for reverse causality influencing the epigenetic ageing phenotypes. The authors should acknowledge this limitation, and ideally, test to what extent reverse causality plays a role in their MR findings using bidirectional MR (or similar methods).

Author response: We agree with the reviewer that reverse causality is a challenge in MR and have added analyses to address this area of concern. Regarding the *cis*-eQTL MR analyses, we are unable to perform bidirectional MR because we cannot create instruments for our aging phenotypes only using SNPs within the locus of a single gene of interest. However, we have added the MR Steiger test of directionality to test for reverse causality in our *cis*-eQTL MR analyses (Supplementary Data 13-17). In only one case does our eQTL instrument explain more variation in an outcome phenotype than in an exposure phenotype (*CD248* on HannumAge, albeit at a nominally significant level), which provides some reassurance that our results do not reflect trait on gene expression effects. We have amended our manuscript in response to the *CD248* on HannumAge Steiger finding in the following ways:

Results (added): *"Moreover, our top gene exposure associated with HannumAge, CD248, may be a product of reverse causality as per the MR Steiger test of directionality (Supplementary Data 14)."*

Results (deleted): *"Increased expression of our top HannumAge-associated gene, CD248, decreased schizophrenia risk, but also increased the prevalence of osteosclerosis and related diseases"*

Table 2, legend (added): *"*This finding may result from reverse causality, as per the MR Steiger test of directionality."*

To address the possibility that our immune system-wide MR on epigenetic age acceleration may suffer from reverse causality, we have made these analyses bidirectional (Supplementary Data 43-47). In no instance did EAA or multivariate longevity significantly affect an immune phenotype at a relaxed FDR threshold of 0.2. In other words, none of the relationships between genetically proxied immune cell traits and EAA were significant in the reverse direction, providing some evidence that the findings from our immune system-wide MR on epigenetic age acceleration do not reflect reverse causality. We describe this finding in the Results as follows:

"To identify immune traits that affect EAA, we used a relaxed FDR threshold of 0.2 because of the relatively lower statistical power of our EAA samples. At this relaxed threshold, CD8 on terminally differentiated CD8+ T cells, CD80 on CD62L+ myeloid dendritic cells, and CD28 on CD28+ CD45RA+ CD8+ T cells were shown to increase IEAA (Table 3, Supplementary Data 38). None of these relationships were significant in

the reverse direction at an FDR of 0.2 (i.e., MR of aging exposures on immune trait outcomes) (instruments contained in Supplementary Data 42, results contained in Supplementary Data 43-47)."

As an additional sensitivity analysis to address the reviewer's concerns, we performed Steiger filtering in our *cis*-eQTL MR analysis and our bidirectional MR analyses of aging traits and immune cell traits. We calculated MR effect estimates using instruments with any SNPs that explained more variance in an outcome than in an exposure removed—these post-Steiger results are displayed in the relevant tables in our Supplementary Data. For our *cis*-eQTL MR analysis, only two of our genetic instruments were impacted by Steiger filtering: the *CD248* instrument on HannumAge (described above) and the *KCNJ14* instrument on GrimAge. We failed to identify a significant relationship between *KCNJ14* and GrimAge both before and after Steiger filtering. For our MR analysis of immune cell traits on aging outcomes, no SNPs were removed from any instrument by Steiger filtering. Finally, while Steiger filtering removed SNPs from several of our instruments in the MR analysis of aging traits on immune cell outcomes, this MR analysis identified zero significant or near-significant relationships.

Despite these steps, we recognize that methods like the MR Steiger test of directionality and bidirectional MR are imperfect tests of reverse causality. Therefore, we have removed the language quoted in the reviewer's comment from the Discussion and added the following language:

"However, describing relationships between EAA and immune-related traits is complicated by the fact that DNA methylation affects cellular differentiation and cell type composition,⁵⁸ making analyses of these traits potentially susceptible to reverse causality."

Minor comments:

4. The word "longevity" in the context of GWAS is understood to refer to an exceptional survival phenotype (PMID 25814633). The authors use of "longevity GWAS", or more misleadingly, "GWAS meta-analysis of longevity", can easily be misinterpreted as a (meta-analysis of multiple) GWAS of exceptional survival. The GWAS in question is actually a multivariate analysis of three distinct longevity-related phenotypes (parental lifespan, healthspan, and longevity). I encourage the authors to consider using more nuanced language when referring to this GWAS throughout their manuscript.

Author response: We appreciate and agree with the reviewer's comment. We now refer to our longevity phenotype as "multivariate longevity." We define multivariate longevity in the following locations in the manuscript.

Abstract: "...a human longevity phenotype comprising healthspan, lifespan, and exceptional longevity ("multivariate longevity")"

Introduction: "...a multivariate, longevity-related phenotype comprising parental lifespan, healthspan, and exceptional longevity (referred to hereafter as "multivariate longevity")"

We detail the phenotypes comprising multivariate longevity in the “Multivariate longevity” section of the Methods.

5. The sample size of the multivariate longevity GWAS is cited to be $N = 1,249,465$, but according to the original article, it should be either $N = 1,349,432$ for total sample size or $N = 709,709$ for the effective sample size (PMID 32678081). Regardless, given this dataset is composed of GWAS with different study designs, it would be more transparent to list the sample size of each study separately rather than summarising the dataset with a single number.

Author response: Thank you for catching this error. Throughout the manuscript, we have replaced “ $N = 1,249,465$ ” with “ $N_{total} = 1,349,432$, $N_{effective} = 709,709$.” Additionally, we have more explicitly stated the sample sizes of phenotypes included in the multivariate longevity phenotype in the “Multivariate longevity” section of the Methods.

6. The authors present a list of TWAS genes they deem “high-confidence”, acknowledging only a few of these genes are significant in multiple ageing-related GWAS. However, looking at TWAS statistics, it appears some of the genes do not even have concordant signs of effect between phenotypes. For example, TWAS Z scores for the newly highlighted FLOT1 gene are HannumAge -5.83 , Longevity 5.09 , GrimAge 4.44 , IEAA 2.39 , PhenoAge -2.16 , (i.e. pro-longevity in HannumAge/PhenoAge/Longevity, and anti-longevity in IEAA/GrimAge). The authors should take these conflicting statistics into account when discussing the reliability of their results.

Author response: We agree with the reviewer that the discordant effect directions we find for certain high-confidence genes like *FLOT1* are somewhat counterintuitive. However, given the high-throughput and hypothesis-free nature of the study, we caution against drawing conclusions from findings that fail to surpass an FDR-corrected significance threshold. Moreover, the forms of epigenetic age acceleration we analyzed in this study may well have different transcriptomic signatures. We note in the Discussion:

“At the transcriptomic level, there was one TWAS finding, FLOT1, shared between EAA (HannumAge) and multivariate longevity, which was only high confidence for HannumAge. Meanwhile, four TWAS associations were associated with multiple measures of EAA, including the high confidence associations of TPMT and NHLRC1 with both IEAA and PhenoAge (Fig. 2). Furthermore, functional analyses of our high confidence genes using GO revealed that, generally, genes implicated in each of our five aging phenotypes had distinct biological functions. The unique transcriptomic signatures of our four EAA phenotypes is notable and logical in the context of previous studies showing that different epigenetic clocks contain generally distinct CpG sites in distinct genomic regions.⁵⁶ We postulate that our four EAA measures may reflect unique epigenetic aging phenomena due to differences in training outcomes and populations.⁵⁷”

Nonetheless, we recognize the validity of the reviewer’s concern and have added the following language to the limitations section of the Discussion:

“The importance of a cautious interpretation of even our high confidence findings is exemplified by the incongruous TWAS effect directions for genes like FLOT1, which is associated with decelerated HannumAge and greater multivariate longevity, but which nominally associates with accelerated IEAA and GrimAge.”

7. While the authors use a suite of standard MR models when performing *cis*-eQTL MR, they only report effect sizes and P values regarding the MR slope. However, it would be useful to see statistics regarding the MR-Egger intercepts in a table, as a significant MR-Egger intercept can indicate the presence of directional horizontal pleiotropy. For the metabolome-wide and cell-type MR analyses, the authors should report MR-Egger intercept standard errors and P values in addition to the MR-Egger intercept estimates which are currently listed in their tables.

Author response: We thank the reviewer for noticing this missing data. We have added MR-Egger intercepts to our *cis*-eQTL MR Supplementary tables (Supplementary Data 13-17). We have also added results from Cochran Q heterogeneity tests to these tables. We see very little evidence of directional pleiotropy or heterogeneity using these tests.

Additionally, we have added MR-Egger standard errors and P values to our metabolome-wide (Supplementary Data 25-29) and immune-system wide (Supplementary Data 37-41, 43-47) MR analyses. In some cases, the MR-Egger statistics read “NA” because MR-Egger is not possible for analyses using instruments comprising fewer than 3 variants. Based on the MR-Egger intercept test, we find almost no evidence for directional horizontal pleiotropy across these MR analyses, except for our analysis of metabolites on GrimAge acceleration. However, this analysis yielded no significant results, so this directional horizontal pleiotropy should not significantly impact the conclusions of our study.

8. Why were MR Steiger tests performed only for exposure-outcome pairs with more than one genetic instrument? It is my understanding the test can be performed with a single SNP as well.

Author response: We apologize for this oversight. We have now performed the MR Steiger test for all exposure-outcome pairs in all MR analyses in our study, including those with single SNP instruments. The results of these tests are included in the rows of their respective exposure-outcome pairs in the Supplementary Data. We summarize the results of these Steiger tests in response to the reviewer’s third comment.

9. Strictly speaking, the MR analysis of metabolite measures on the multivariate longevity GWAS is not a two-sample analysis: UK Biobank samples were used both in selecting the exposure instruments and in calculating two of the three outcome GWAS statistics. The authors should acknowledge sample overlap will bias MR effect sizes towards the observational correlation between exposures and longevity, rather than the null (increasing false positive findings). In addition, the authors adjust for multiple testing of metabolites using FDR correction, yet this correction assumes each hypothesis test is independent. As some metabolite traits are ratios of other traits, this independence assumption clearly does not hold. A more appropriate method to correct for multiple testing would be to estimate

the number of independent metabolite traits (using e.g. principal component analysis) and adjust for this estimated number instead.

Author response: In response to the reviewer’s point about sample overlap, we have included the following language in the limitations section of the Discussion:

“Moreover, the GWASs of circulating metabolites and two of the three multivariate longevity phenotypes—healthspan and lifespan—included participants from the UK Biobank. MR analyses using these data therefore had sample overlap, which may have biased effect estimates toward their observational associations and away from the null,⁶⁶ although recent literature suggests that two-sample MR methods can be safely used for one-sample MR in large biobanks.⁶⁷”

Regarding the reviewer’s point about FDR correction, we have amended our multiple testing correction for our metabolite-on-aging MR analyses as follows, based on the principal component analysis described in this link:

Methods: “To define significance, we divided the conventional significance threshold of $P < 0.05$ by 41 principal components that explain 99% of variation in the levels of circulating metabolites, yielding a Bonferroni-corrected threshold of $P < 0.00122$ (<https://nightingalehealth.github.io>).”

Using this new significance threshold, there remain zero significant metabolite associations with EAA, while 160/249 metabolites have significant associations with multivariate longevity.

Finally, we similarly amended our multiple testing corrections for our MR analyses of immune system traits on aging outcomes and our new MR analysis of aging traits on immune system outcomes based on the number of independent GWAS traits in Orru et al.’s dataset:

Methods: “For both sets of MR analyses of the immune system, we defined significance using FDR thresholds based on 539 independent immune cell traits,²⁹ calculated separately for each aging outcome.”

This change did not substantially change the results of our immune trait MR analyses.

10. The authors have a section on computational drug repurposing and drug set enrichment, but only list its methods and results in a supplementary file without any mention in the main text. As these analyses are not central to the current study and suffer from the same enrichment bias due to correlated TWAS statistics (see concern 1.1 above), I politely question why this section was included in the manuscript. The authors should either list and discuss (a summary of) these supplementary results in the main text or remove them from the study.

Author response: Thank you. Given the limitations of these analyses and to enhance the focus of the manuscript, we have removed computational drug repurposing and drug set enrichment from the paper.

To further focus the paper, we now restrict our phenome-wide association studies to genes identified as significant in our *cis*-eQTL MR analyses.

Finally, we have removed our query of the Drug Gene Interaction Database for drugs that targeted our *cis*-eQTL MR-identified genes. We felt that this analysis was not insightful enough to warrant inclusion in the manuscript.

11. Please specify whether the heritability estimates cited in the introduction are SNP-based or pedigree-based, and double-check the numbers cited are correct. From my understanding, the lower bound of SNP-based heritability should be 0.10, as estimated for GrimAge acceleration in McCartney et al. (PMID 34187551; Table S19), whereas the upper bound of pedigree-based heritability estimates should be 0.43 (PMID 25633388).

Author response: The heritability estimates of 0.15 to 0.19 specified in our original submission were SNP-based and were those cited in the introduction of McCartney et al., but we agree with the reviewer that these estimates don't reflect the entire range of heritability estimates for epigenetic age acceleration. As per the reviewer's suggestion, we have revised our description of the heritability of epigenetic age acceleration to the following:

“Additionally, epigenetic clocks are influenced by genetic factors and have heritability estimates ranging from 0.10 (single nucleotide polymorphism (SNP)-based heritability of GrimAge acceleration) to 0.43 (pedigree-based heritability of accelerated Horvath and Hannum clocks).^{10,11}”

12. In Supplementary Tables 1-5, any TWAS results with $P > 0.05$ have permutation P values of zero. If these TWAS results were not tested using the permutation method, these statistics should be set to NA rather than zero to avoid confusion.

Author response: We thank the reviewer for identifying this potential point of confusion. We did not test TWAS results with $P > 0.05$ using the permutation method. We have amended the Supplementary Data as suggested.

13. Figure 2 and Figure 3, which contain the results from metabolome-wide MR and cell-type enrichment respectively, do not appear to be very informative. I would prefer the authors focus on a summary of the significant MR effects, rather than displaying the full results for all longevity traits and metabolites/cell-types. For example, in the metabolome MR, the effects of cholesterol-to-lipids ratio traits are largely captured by cholesterol and lipid traits, and the only significant MR effects for these categories are on the multivariable longevity phenotype. The authors may want to consider an alternative way to highlight the most pertinent findings from these analyses.

Author response: We agree that these figures are not as informative as they could be. To address this issue, we have replaced the metabolome-wide MR figure (now Figure 3) with a volcano plot that only focuses on the metabolite effects on the multivariate longevity phenotype. The y-axis of the plot displays Z-scores, as the beta values of the metabolic phenotypes cannot all be converted into the same unit. We label the eight most significant metabolites with positive Z-scores and the eight most significant metabolites with negative Z-scores to highlight these findings. We still include the ratio traits in our figure because we believe it is important to

highlight composite metabolic phenotypes that may have a greater impact than their component traits due to interactions between these traits.

We have amended the cell-type enrichment analysis figure (now Figure 4) to be a four-panel bar plot displaying only the significant cell-types for each of the four epigenetic age acceleration traits. We excluded the multivariate longevity phenotype from this figure because no cell-types were significantly associated with it. We have also created ten new supplementary figures (Supplementary Figures 1-10), each displaying the results of a CELLECT-MAGMA or CELLECT-LDSC analysis of a single aging phenotype. These figures are not convoluted—they contain a single analysis of a single phenotype—but they still illustrate the overall enrichment of genes associated with EAA (and to a lesser extent, multivariate longevity) in immune cells and immune cell precursors.

We believe that these changes make our figures more intuitive and highlight our most pertinent findings.

14. Reference 6 and 48 refer to the preprint and published versions of the same Gibson et al. paper. Ditto for reference 9 and 15 for the McCartney et al. paper. The published versions of these studies should be cited, unless there is a good reason to also cite the preprints.

We appreciate that the reviewer noticed these duplicate citations. We have removed all duplicate citations and replaced citations of preprints with citations of published papers, whenever possible.

Reviewer #2 (Remarks to the Author):

Mavromatis and colleagues conducted a comprehensive multi-omic analysis to address the underlying mechanisms of aging and to identify druggable molecular targets (TPMT is highly relevant). Leveraging large publicly available datasets and computational resources, the study integrated genome, transcriptome, and metabolomics and identified new genes for epigenetic age acceleration and longevity. The results highlight the important roles of immune and inflammation in the aging process. The study is timely and the findings, if replicated, may add new knowledge for aging-related diseases.

Comments

1. Identifying highly confident genes using FOCUS is a strength of the study. Some high confidence genes by TWAS are relevant to aging based on previous research. However, it is puzzling that no gene is shared among the four EAA phenotypes measured by epigenetic clocks, and only one gene overlapped with longevity. This raises the question of whether the four clocks measured the same trait. I would appreciate elaboration or further discussion of this question.

Author response: We agree with the reviewer that this is an important finding—the epigenetic clocks analyzed generally do not have convergent TWAS or FOCUS findings. We have

modified the Discussion to include a paragraph elaborating on the lack of convergence among high confidence genes associated with different aging phenotypes:

“At the transcriptomic level, there was one TWAS finding, FLOT1, shared between EAA (HannumAge) and multivariate longevity, which was only high confidence for HannumAge. Meanwhile, four TWAS associations were associated with multiple measures of EAA, including the high confidence associations of TPMT and NHLRC1 with both IEAA and PhenoAge (Fig. 2). Furthermore, functional analyses of our high confidence genes using GO revealed that, generally, genes implicated in each of our five aging phenotypes had distinct biological functions. The unique transcriptomic signatures of our four EAA phenotypes is notable and logical in the context of previous studies showing that different epigenetic clocks contain generally distinct CpG sites in distinct genomic regions.⁵⁶ We postulate that our four EAA measures may reflect unique epigenetic aging phenomena due to differences in training outcomes and populations.⁵⁷ However, we note that high confidence genes linked to all five of our aging outcomes were associated with functions broadly related to insulin signaling, mitochondrial function, and metabolism. These biological domains are likely to play roles in diverse aging-related phenotypes.”

2. Although the positional gene set of chr6p21 is common among the three traits, less than 50% of the significant pathways are shared across different measures of aging and longevity. Thus, these discrepancies should be discussed. The region of chr6p21 has been linked to many pathological conditions so it is not surprise to see the enrichment with mortality and longevity.

Author response: In response to Reviewer 1, we have replaced our GSEA of TWAS summary statistics with a follow-up analysis focusing only on the 28 “high confidence” genes that were associated with the epigenetic clocks and multivariate longevity in the FUSION TWAS pipeline (transcriptomic imputation + fine mapping + conditional analysis). We made this change to better focus our manuscript on high confidence associations and improve our characterization of these genes’ biological functions. Additionally, as Reviewer 1 emphasized, our previous GSEA was likely overstating positional gene set enrichment because TWAS is known to prioritize clusters of neighboring genes in linkage disequilibrium with one another (“enrichment bias”).

In the aforementioned new analysis, we used Gene Ontology datasets to functionally annotate our high confidence genes but did not include the positional data to avoid enrichment bias. Therefore, we have removed the chr6p21 GSEA findings from our manuscript. However, as you requested, we have enhanced our discussion of the discrepancies and similarities in pathway-level biology between our five aging traits in the Discussion paragraph quoted in response to the reviewer’s first comment and in the Results, as follows:

“With the exception of IEAA and PhenoAge, which share high confidence associations with TPMT and NHLRC1, we observed relatively limited overlap in the biological functions of genes associated with different aging traits (Supplementary Data 11). The few functions common to EAA and multivariate longevity genes included “negative

regulation of gene expression” and “positive regulation of transcription, DNA-templated.” Yet, while few specific GO gene sets were shared by genes associated with different aging traits, genes associated with all five traits had functions broadly related to insulin signaling, mitochondrial function and metabolism (Supplementary Data 11).”

3. The imputation of the transcriptome is built on the previous large GWAS on EAA by McCartney et al., which reported 137 significant loci. What proportion of the loci is shared between TWAS in this analysis and the GWAS on EAA? The significance of the common genes or eQTLs should be examined.

Author response: Generally, most genes identified in a TWAS study are in the same locus as significant SNP in that TWAS’s source GWAS (e.g., PMID: 31230729, PMID: 29915430, PMID: 29632383). In the case of our study, 20/23 of the high-confidence genes identified in our TWAS of EAA are within 500 kilobases of a lead SNP identified by McCartney et al., the traditional window designating shared signals between TWAS and GWAS (PMID: 31230729, PMID: 29915430, PMID: 29632383). In response to the reviewer’s request, we have now explicitly stated this in the results section:

“Thus, in total, 19/22 of our high confidence findings for EAA and 7/7 of our high confidence findings for multivariate longevity likely reflect signals from their respective source GWAS.”

We discuss the significance of several of the shared GWAS/TWAS findings in the discussion (e.g., *TMPT*, *TOMM40*, *SESNI*). We also emphasize the TWAS findings that do not appear in McCartney et al. (*FLOT1*, *KPNA4*, *TMX2*) because of these genes’ novelty.

4. In Figure 3, many metabolites from the MR results for epigenetic age and longevity illustrate different directions. Please explain this discrepancy.

Author response: We appreciate the Reviewer’s feedback. Firstly, it is important to note that an increase in epigenetic age acceleration should, theoretically, correspond to a decrease in multivariate longevity. In other words, metabolites that make you age faster (positive effect direction on EAA) should decrease healthspan, parental lifespan, and exceptional longevity (negative effect direction on multivariate longevity). We have added an explicit statement to this effect in the “Effect directions and sizes” section of the Methods:

“Throughout this study, positive effects on multivariate longevity and negative effects on EAA should be interpreted as beneficial to human health.”

Additionally, some metabolites should be expected to have opposite directions of effect on our aging-related phenotypes (e.g., LDL-C and HDL-C).

Finally, given your and the other Reviewers’ feedback, we have recreated Figure 3 such that it only displays the multivariate longevity phenotype. None of the analyzed metabolites had statistically significant effects on EAA phenotypes, and the inclusion of these effect estimates in the figure encouraged misleading comparisons.

We hope that these comments and changes make our metabolome-wide MR results clearer.

5. A previous study reported that TERT is associated with IEAA and EEAA. However, this gene is not replicated in this large study.

Author response: We thank the Reviewer for pointing out this interesting genetic finding. While *TERT* is identified in our source GWAS of epigenetic age acceleration (McCartney et al.) and a previous GWAS of IEAA (Lu et al.), we fail to identify it in our TWAS. We believe this relates to our cross-tissue gene expression weight dataset. To address this important discrepancy, we have added the following language to our Discussion:

“Additionally, our Genotype-Tissue Expression Project version 8 (GTEx v8) cross-tissue gene expression weight dataset did not include expression weights for all genes that may be relevant to epigenetic aging or longevity. For instance, TERT, which was identified in GWASs of EAA^{10,68} and the overexpression of which increases IEAA in primary human fibroblasts,⁶⁸ was not contained within this reference data.”

Furthermore, in our *cis*-instrument MR analysis of *TERT* expression on four epigenetic age acceleration outcomes, we fail to identify significant relationships, although, as in Lu et al., the directionality of our effect estimates of *TERT* expression on each of our outcomes suggests that greater *TERT* expression predicts greater epigenetic age acceleration (Supplementary Data 13-16). Because our *TERT* instrument only includes one SNP, it may be underpowered to detect an effect.”

6. Importantly, besides European ancestry samples, 6,482 African American individuals were included in the McCartney’s paper. It would be helpful to report the TWAS findings and other similar analyses in AA ancestry samples so the results may be more generalizable, especially for identifying the druggable genes.

Author response: We appreciate the Reviewer’s suggestion. Increasing the diversity of “omic” studies is crucial in producing more generalizable science and medicine and avoiding the perpetuation of health disparities (PMID: 30901543). Unfortunately, there are few large and comprehensive data sources available in non-European ancestry populations, and particularly few in African ancestry populations, complicating this effort (PMID: 30901543). Commendably, McCartney et al. perform a GWAS of epigenetic age acceleration in 6,482 African American (AA) individuals. However, extending our TWAS to an AA sample would require cross-tissue eQTL weights derived from an AA sample, as previous research has shown that eQTL mapping has limited generalizability across ancestry groups (PMID: 32797036) due to ancestry-based differences in linkage disequilibrium structure (PMID: 30901543) and allele frequencies (PMID: 30096133). Unfortunately, few eQTL mapping studies have been conducted in AA individuals, and those that have are limited in size and focus on one cell type (PMID: 30096133, PMID: 32220292). These methodological and data availability issues make extending our cross-tissue transcriptomic analyses to an AA sample unfeasible.

Diversifying our other analyses is impeded by similar data availability issues. For instance, the GWAS we used to conduct our analysis of 731 immune cell traits, Orru et al., was conducted in a Sardinian sample, while the Nightingale metabolomics GWAS data we used to conduct our

analysis of 249 circulating metabolites was conducted in a predominantly White British sample. Because the genetic architecture of complex traits has limited generalizability across ancestry groups (PMID: 30901543), performing analyses in AA samples using these European ancestry-derived genetic instruments would not be methodologically sound.

While we are unable to perform our analyses in an AA sample, we believe that the point the Reviewer makes is very important. Thus, we have added an enhanced discussion of the limited generalizability of our study to the limitations section of the Discussion:

“Also, due to a paucity of diverse “omic” datasets, the human datasets we used throughout this study almost exclusively comprise participants of European ancestry. Due to differences in allele frequency, LD structure, and the genetic architecture of complex traits between ancestral populations, multi-omic analyses have limited trans-ancestral generalizability.⁶⁴ More comprehensive genomic data on health-related traits, gene expression, circulating metabolites, and immune phenotypes would allow studies like this one to be performed in non-European ancestry populations, avoiding the perpetuation of health disparities produced by ungeneralizable science.⁶⁵”

Reviewer #3 (Remarks to the Author):

I really struggled to follow this paper. Maybe I am not the intended audience for this manuscript, but I didn't take much from this work - for me, it was a disjointed set of analyses with no obvious prior hypothesis, and no strong conclusion. I appreciate that this hypothesis-free discovery approach is one way to do science and maybe there are others for whom these findings are interesting - maybe for some, this is a goldmine or treasure trove of information. So please feel free to ignore this review if those who are part of the intended audience appreciate it, but honestly - I doubt if I'll remember anything from reading this paper even in a week's time.

Author response: To address the Reviewer's broader concern that the manuscript is disjointed and unfocused, we have made some changes to remove superfluous analyses. Specifically, we removed the computational drug repurposing and drug set enrichment analyses that appeared in the Supplement, the Drug Gene Interaction Database (DGIdb) analysis that follow-up our MR analysis of the druggable genome (in which we used DGIdb to identify drugs that targeted the genes we identified with MR), and we restricted our phenome-wide association studies to genes that were significant findings in our MR analysis of the druggable genome.

We have also, as the Reviewer requested in his/her comments, significantly amended our methods section to make it more comprehensible and altered our introduction to better motivate our study. Finally, we have recreated the figures for our metabolome-wide MR (Figure 3) and our cell-type enrichment analysis (Figure 4) so that they are easier to interpret.

We believe that these changes should make the manuscript more accessible.

We believe that our hypotheses-free approach offers important advantages. Given the dearth of knowledge about the biological mechanisms underlying aging, and the biological “meaning” of

epigenetic clocks in particular, we felt that restricting our study to a subset of genes, metabolites, or immune cell traits would be inappropriate, especially given the wealth of “omic” data at our disposal. Moreover, hypothesis generation is, in our view, the core purpose of large-scale multi-omic studies, and we hope the comprehensiveness of our data will make it a valuable source of hypotheses for scientists exploring broad questions related to the biology of healthy and pathological aging. For example, to follow up on our results, researchers may investigate the hypothesis that lipid-lowering drugs, which have broad effects on the circulating metabolome, increase longevity without impacting epigenetic or biological aging processes. Or, scientists may explore the relationship between genes in the 6p22.3 locus (i.e., *TPMT* and *NHLRC1*) and aging.

We hope this response clarifies why we believe our study’s approach is valuable.

Some comments:

1. I appreciate that in places the authors have tried hard to make their work understandable, but in other places, the approach taken and the motivation for this was unclear. As an example, the first substantive sentence of the methods reads: "Our TWAS analyses used pre-computed, cross-tissue, single nucleotide polymorphism (SNP) expression quantitative trait loci (eQTL) weights generated using sparse canonical correlation analysis (sCCA)." While I understand most of these words individually, their combination is beyond me. An explanatory phrase to motivate would help, eg: "To summarize the data at each gene region into a single variable, our TWAS analyses...". Or whatever the motivation for this is. Please remember - the point of a scientific paper is not to dazzle the reader with your ability to form complex sentences, but to communicate, and to be understood.

Author response: We appreciate the Reviewer’s comment about the comprehensibility of our Methods section. In response, we have gone through the Methods section and rewritten confusingly worded sentences and phrases. We have also attempted to reduce unnecessary jargon and have done some reorganization. Moreover, we have added sentences motivating our methods, as suggested by the Reviewer, where applicable. For example, we changed the sentence that the Reviewer quoted in his/her comment to state:

“In our TWAS analyses, we sought to translate SNP associations with our aging outcomes into gene transcript associations with these outcomes in a tissue nonspecific manner to capture the cross-tissue nature of aging. Therefore, we used cross-tissue gene expression weights generated with sCCA.¹⁹ These sCCA gene expression weights were derived from GTEx v8 atlas of eQTLs.⁷⁴”

We hope that our revamped Methods section is clearer and better motivated.

2. What do the points represent in Figure 3? I don't understand this figure - why is there more than one point per metabolite/outcome pair?

Author response: We agree that the figure displaying the results of our metabolome-wide MR analysis was previously confusing. In the previous figure, we grouped our metabolites by

metabolite category and plotted the effect of each metabolite on all five EAA/longevity phenotypes, using color-coding to indicate which outcome a given point corresponded to.

We have since recreated Figure 3 as a volcano plot, displaying P-values on the x-axis and Z-scores on the y-axis. In our recreated figure, we only display the effect estimates on our multivariate longevity phenotype; no metabolites had significant effects on any of our four epigenetic age acceleration phenotypes, and this change makes the figure less cluttered. Unlike traditional volcano plots, we plot Z-scores, rather than beta values, because it is impossible to standardize our beta values to a single unit since some of our metabolic phenotypes are concentrations and others are unitless ratios. Finally, we label the eight most significant metabolic phenotypes with positive Z-scores and eight most significant metabolic phenotypes with negative Z-scores to emphasize important findings. We believe these changes make Figure 3 significantly clearer and more informative.

3. Did you consider colocalization for the top drug-target hits?

Author response: Thank you for the idea. In response to your comment and the comment of another Reviewer, we have added colocalization, using the coloc package in R, to assess our significant drug-target MR (Supplementary Data 13-17) and TWAS hits (Supplementary Data 1-5). We describe our colocalization analyses as follows in the Methods section of the manuscript:

Post-TWAS colocalization: *“Finally, in line with previous TWAS studies,^{76,77} we performed colocalization of our TWAS-significant genes using the coloc R package (version 5.1.0.1),²⁰ implemented in FUSION. This allowed us to assess the probability that our TWAS associations reflected linkage between distinct causal SNPs (PP.H3) or a single causal SNP (PP.H4). We defined PP.H4 > 0.75 as strong evidence of colocalization.”*

Post-MR colocalization: *“Findings remaining significant after this FDR correction were subject to colocalization analysis, which used the same methods²⁰ and interpretation as our post-TWAS colocalization analysis.”*

Most of our TWAS findings showed strong evidence of colocalization (19/28 IEAA-associated genes, 8/20 for HannumAge, 4/4 for GrimAge, 4/7 for PhenoAge, 21/34 for multivariate). By contrast, many of our MR findings failed to show strong evidence of colocalization (0/4 IEAA-associated genes, 1/6 for HannumAge, 1/2 for PhenoAge, 6/27 for multivariate longevity). We describe these findings in the Results as follows:

Post-TWAS colocalization: *“Most of these features colocalized with their respective aging phenotype, suggesting that one or more shared, pleiotropic SNP influences both gene expression and said aging phenotype (19/28 for IEAA, 8/20 for HannumAge, 4/4 for GrimAge, 4/7 for PhenoAge, 21/34 for multivariate longevity).”*

Post-MR colocalization: *“However, few of the genes identified by the drug-target MR analysis, including neither TPMT, C4B, nor NHLRC1, demonstrated strong evidence of colocalization with our aging phenotypes (0/4 IEAA genes colocalized, 1/6 HannumAge genes, 1/2 PhenoAge genes, 6/27 multivariate longevity genes) (Supplementary Data 13-17).”*

We discuss the implications of our colocalization findings with respect to two of our high confidence gene findings, *TPMT* and *NHLRC1*, in the Discussion:

“Although TPMT was prioritized by our transcriptomic analyses, it is important to note that colocalization results for our TWAS and drug-target MR analyses of TPMT yielded no evidence for a causal relationship between TPMT and EAA. TPMT resides in the 6p22.3 locus next to NHLRC1, a gene with high-confidence TWAS associations with IEAA and PhenoAge and which significantly decelerated IEAA and nominally decelerated PhenoAge, according to our MR results. NHLRC1 encodes malin, a ubiquitin ligase involved the regulation of glycogen.⁴⁷ NHLRC1 showed strong evidence of colocalization with IEAA and PhenoAge in TWAS follow-up analyses but failed to show evidence of colocalization with these traits in MR follow-up analyses. Because of the mixed evidence linking TPMT and NHLRC1 to EAA outcomes, functional, fine-mapping, and interventional studies should investigate these genes and their role in aging.”

4. Everything is done in such a high-throughput way that it's difficult to focus on any one analysis, but I was slightly concerned about the MR methods that exclude variants from the analysis (MR-Lasso and penalized weighted median). If this method is excluding 1 or 2 variants out of 10-20, then that's fair enough. But if it is excluding a substantial fraction of the variants from the analysis, then it may give a false impression of consistency in the analysis - everything is consistent if you remove enough of the heterogeneity. I appreciate this isn't practical in each case, but it'd be nice to know what % of variants are being excluded from the analysis for the key metabolites.

Author response: We appreciate this suggestion. We have added a column in Supplementary Data 29 that quantifies the percentage of variants excluded in the post-Lasso IVW estimate of each metabolite on longevity. The average percentage of variants excluded by Lasso is 31.04%. Although this percentage is substantial, the beta values and top metabolites we report using pre- and post-Lasso IVW are quite similar (Supplementary Data 29), suggesting that our Lasso correction is not drastically altering our results. Moreover, the results of a simulation study (PMID: 32249995) suggest that MR-Lasso performs well, both in terms of bias and precision, when 30% of variants included in a genetic instrument of 10, 30, or 100 variants are invalid instruments.

We did not use Lasso to exclude variants for our MR analysis of circulating metabolites on epigenetic age acceleration because we did not detect substantial heterogeneity in these estimates with the Cochran Q test, so this concern does not apply to these analyses.

Additionally, to standardize the methods used between our metabolome-wide MR analysis and our *cis*-eQTL MR analysis, we have removed the penalized weighted median estimator from our Supplementary Data. We now only report IVW, MR Egger, weighted median, and weighted mode estimators.

5. My final comment is a somewhat broad and probably unhelpful comment about ageing as an outcome. If you are developing a drug to lower blood pressure, then everything is simple - you take the drug, and measure blood pressure in a week - did it go down or not? Does the drug work? - you should get a definitive answer in quick order. If you are

developing a drug to slow the ageing process, then validating the drug is much harder. And if you can't validate the drug, then what is the point in investigating the mechanisms? I realise this is an unfair question, as scientific research shouldn't only be undertaken because it is practically useful. But I did wonder if you could comment on the relative value of an investigation into longevity as opposed to an investigation into the common components of longevity (eg cardiovascular disease, cancer, dementia), which are more likely to have a direct biological mechanism, as well as a potential target population for any drug developed.

Author response: We understand the Reviewer's concerns and agree that validating a drug aimed at slowing the aging process is currently very difficult due to the long, expensive clinical trials that would be required to identify significant group differences in life expectancy and age-related disease onset. Drugs that protect against cardiovascular disease are much easier to develop, because, like the Reviewer mentioned, there are extremely well-validated and well-understood biomarkers, like blood pressure and LDL cholesterol, that can be used to guide drug development. For aging, there are no such biomarkers. However, epigenetic clocks are considered to be the most promising biological aging biomarker (PMID: 28396265), and there have been recent clinical trials aiming to slow these clocks with pharmacological and lifestyle interventions (PMID: 31496122).

Therefore, improving our understanding of the transcriptomic, metabolomic, and immune-level correlates of these clocks, and comparing them to the biological correlates of the clinically relevant measures of longevity comprising our multivariate longevity phenotype, is important for advancing our understanding of these clocks and of the potential for them as biological targets for the aging process. By furthering our understanding of EAA and revealing its similarities to and differences from multivariate longevity, we can inch closer toward the possibility of anti-aging drug development. This is important because, compared to the development of drugs that combat a single disease, the development of drugs that even moderately slow the biological processes linking age to functional and health decline could have a near universal target population and could lead to enormous economic and health benefits (<https://doi.org/10.1038/s43587-021-00080-0>).

In response to the Reviewer's thought-provoking comment, we have revamped paragraphs one and two of our Introduction to include the study motivation described above:

“Aging is often accompanied by a loss of independence, disability, and the onset of diseases like cancer, cardiovascular disease, and neurodegenerative diseases which cumulatively represent the leading cause of death in developed nations.¹ Traditionally, medical interventions seek to delay the onset of, cure, or treat the symptoms of individual age-related diseases. However, recently, describing and pharmacologically targeting the biological processes linking age to functional and health decline has received attention as an alternative strategy for increasing healthy years-lived.^{2,3} This strategy could offer health and economic gains that substantially outweigh those achieved by targeting specific diseases.³ However, the feasibility of slowing biological aging is currently limited. Long, expensive clinical trials would be required to identify anti-aging interventions that increase life expectancy in healthy individuals.⁴ Moreover, while promising drug candidates for specific diseases can be prioritized and even approved by

regulators if they alter well-validated disease biomarkers (e.g., low-density lipoprotein cholesterol and cardiovascular disease), biomarkers for biological aging are poorly understood and validated.

Epigenetic clocks, which incorporate data about the methylation status of CpG sites across the human genome into weighted linear equations to predict chronological age and/or age-related disease onset,⁵ are generally considered to be the most promising biomarker for biological aging,⁶ and there have been several recent clinical trials aiming to slow epigenetic clocks with pharmacological and lifestyle interventions.^{7,8} These clocks show strong correlations with chronological age and other aging-related phenotypes.⁹ Additionally, epigenetic clocks are influenced by genetic factors and have heritability estimates ranging from 0.10 (single nucleotide polymorphism (SNP)-based heritability of GrimAge acceleration) to 0.43 (pedigree-based heritability of accelerated Horvath and Hannum clocks).^{10,11} For some individuals, epigenetic age outpaces chronological age in what is referred to as “epigenetic age acceleration” (EAA). EAA is associated with a number of health conditions and age-related diseases, including substance use behaviors,¹² atherosclerosis,¹³ cancer,¹⁴ and mortality.¹⁵ Given the potential of EAA as an aging biomarker and the large benefits that could be achieved through interventions that slow biological aging processes, it is crucial to describe the biological correlates of EAA, identify targetable biomolecules that modify EAA, and assess the biological similarities and differences between EAA and clinically relevant aging phenotypes like healthspan, lifespan, and exceptional longevity.”

REVIEWERS' COMMENTS

Reviewer #1 (Remarks to the Author):

Thank you for taking the time to respond to my comments. The manuscript has improved considerably since the last version, and all of my initial concerns have been adequately addressed.

PrismEXP is an elegant solution to address my concern regarding inflated GSEA statistics, and the authors' bidirectional MR and Steiger tests have strengthened the argument against reverse causality. The authors' new colocalisation analyses have revealed several of their findings may have been confounded due to genetic linkage. In light of these new findings, the authors have expanded their discussion to moderate their claims of causality for TPMT and NHLRC1.

I have the following remaining comments:

1. The authors currently report colocalisation results in the main text in addition to their previous findings. Lack of evidence for colocalisation indicates the genetic variants underlying gene expression may be distinct from those underlying the ageing-related traits, and therefore, no claims about causality can be made. As such, I suspect readability of the manuscript could be improved if the authors focus on highlighting the results with evidence for colocalisation only (in both the main text and tables).
2. When testing for colocalisation, the default coloc software assumes there is a single causal variant in the region of interest. However, this assumption is likely violated for many genes of interest (for example, many druggable genes have multiple independent cis-eQTL which are used as genetic instruments in MR). There are several extensions to coloc that can take into account multiple causal variants, but it is unclear from the Methods if any of those extensions were used. If all colocalisation results are based on the single variant assumption, these results may be overly conservative.
3. Line 271-273 "Our study also identifies potential druggable genes, including genes located at chromosome 6p22.3 (TPMT and NHLRC1) that could promote human longevity." I believe the wording here is somewhat too strong considering there was mixed evidence for causality of TPMT and NHLRC1.
4. What does the column "In credible set" in Supplementary Data 7 refer to? If it is an indicator for which genes are included in the credible set, why is this value zero for some genes that have a posterior inclusion probability of one?
5. Lines 157-159: "genes associated with all five traits had functions broadly related to insulin signaling, mitochondrial function and metabolism (Supplementary Data 11)." As far as I can tell there are no high confidence genes associated with all five traits. The authors mention the only genes associated with more than one trait are TPMT and NHLRC1 (which are associated with PhenoAge and IEAA).
6. Line 169: "Table 3" should be "Table 2" as the preceding sentence reports drug target MR results.

Reviewer #2 (Remarks to the Author):

The authors have made a careful and thorough revision addressing my concerns. I have a minor comment regarding the little converging signals among different clocks.

While the revision did recognize the issue, it is still unclear what implications these findings are on each clock. Given that five clocks have been extensively used for many studies, I hope the authors will make further efforts to discuss this topic in more detail.

March 17th, 2023

Dear Reviewers,

Thank you again for taking the time to review our manuscript, “Multi-omic underpinnings of epigenetic aging and human longevity.” Because of your help, we believe this version of the manuscript is the strongest yet and that we have effectively addressed lingering methodological and language issues. Please find our point-by-point responses to your comments below.

Sincerely,

Falk W. Lohoff, M.D.

Reviewer #1 (Remarks to the Author):

Thank you for taking the time to respond to my comments. The manuscript has improved considerably since the last version, and all of my initial concerns have been adequately addressed.

PrismEXP is an elegant solution to address my concern regarding inflated GSEA statistics, and the authors’ bidirectional MR and Steiger tests have strengthened the argument against reverse causality. The authors’ new colocalisation analyses have revealed several of their findings may have been confounded due to genetic linkage. In light of these new findings, the authors have expanded their discussion to moderate their claims of causality for TPMT and NHLRC1.

I have the following remaining comments:

1. The authors currently report colocalisation results in the main text in addition to their previous findings. Lack of evidence for colocalisation indicates the genetic variants underlying gene expression may be distinct from those underlying the ageing-related traits, and therefore, no claims about causality can be made. As such, I suspect readability of the manuscript could be improved if the authors focus on highlighting the results with evidence for colocalisation only (in both the main text and tables).

Author response: We appreciate this comment. In accordance with the reviewer’s request, we have focused on highlighting the cis-eQTL MR results with evidence for colocalization in a few ways. First, we have removed from Table 2 the results that do not have strong evidence of colocalization (now assessed with the SuSiE extension of coloc; see response to comment 2). To downplay results without strong evidence of colocalization and emphasize those with strong evidence, we have also altered the main text as follows:

Abstract: We have removed the reference to *C4B* from the abstract because this gene lacked evidence of colocalization.

Results: “While our drug-target MR identified many significant associations, many of these associations, including the associations involving *C4B*, failed to show strong evidence of colocalization (1/4 IEAA genes colocalized, 2/6 HannumAge genes, 2/2 PhenoAge genes, 7/27 multivariate longevity genes) (Supplementary Data 13-17). Non-colocalized gene-trait associations cannot be interpreted as causal relationships. Moreover, our gene most significantly associated with HannumAge, *CD248*, may be associated due to reverse causality, as per the MR Steiger test of directionality (Supplementary Data 14). Despite an overall lack of colocalization, *TPMT* and PhenoAge, as well as *NHLRC1* and IEAA, did show

strong evidence of colocalization and thus may reflect causal relationships. Table 2 displays all colocalized drug-target MR findings.”

Results (bold removed): “SNPs near *C4B*, our other drug target significant for two phenotypes, most prominently associated with reduced risk type 1 diabetes and other autoimmune diseases”

Results (bold removed): “Ultimately, follow up studies should investigate the effects of modulating transcript levels of *TPMT*, *NHLRC1*, *C4B*, and other genes identified in this MR on biological aging.

Discussion: Paragraphs 3-5, which are omitted from this document for brevity’s sake, are significantly altered to focus only on MR findings that colocalized.

We have opted to continue to discuss TWAS results without strong evidence for colocalization in our manuscript and we continue to display these results in Table 1. We believe this is warranted because, as the reviewer mentions in comment 2, the default coloc software that we use, and that is integrated into the FUSION code pipeline, assumes that each gene has a single causal cis-eQTL, which is a conservative assumption that is likely violated in many cases. For our cis-eQTL MR analysis, we are now able to circumvent this single causal variant assumption by using the SuSiE extension of coloc (PMID: 34587156), which does not rely on the single causal variant assumption and should perform well in situations in which multiple causal cis-eQTLs exist for a gene. For our TWAS, we did not perform coloc SuSiE because 1) it cannot currently be implemented in the FUSION pipeline, and 2) we used the FOCUS fine-mapping software, which similarly performs well in situations in which multiple cis-eQTLs exist for a gene (PMID: 30926970) and is the conventional Bayesian fine-mapping approach for TWAS results. We have added language reflecting this aspect of FOCUS to the Methods:

Methods: “Importantly, unlike traditional colocalization, FOCUS performs well in scenarios in which, within a locus, multiple causal variants exist for a given gene, or multiple causal genes exist for a given trait.²³”

To further support our reasoning, we added the following statement in the TWAS Methods subsection “Conditional analyses, permutation testing, and colocalization.”

Methods: “Because traditional colocalization assumes that a single causal variant exists for a given trait in a specified genomic region,²² an assumption that is conservative and likely to be violated,⁸⁰ we used it as a supplementary sensitivity analysis.”

2. When testing for colocalisation, the default coloc software assumes there is a single causal variant in the region of interest. However, this assumption is likely violated for many genes of interest (for example, many druggable genes have multiple independent cis-eQTL which are used as genetic instruments in MR). There are several extensions to coloc that can take into account multiple causal variants, but it is unclear from the Methods if any of those extensions were used. If all colocalisation results are based on the single variant assumption, these results may be overly conservative.

Author response: Thank you for mentioning this important assumptions of the default coloc, which we agree is likely violated for many of our genes. We have run an additional colocalization follow-up to our cis-eQTL MR analysis using the colocalization extension SuSiE (PMID: 34587156), which does not rely on the single causal variant assumption and performs better in situations in which multiple causal cis-eQTLs exist for a gene. We now report our SuSiE findings as our primary colocalization results (see:

Methods → Drug-target Mendelian randomization, Results → MR identifies druggable genes that impact aging traits, Table 2, Supplementary Data 13-17). As described in the aforementioned Results section, we indeed find that the SuSiE extension of coloc identifies more colocated genes than the default coloc. We still report results from the default coloc analysis in Supplementary Data 13-17, but this is merely as an additional sensitivity analysis and is not the colocalization analysis from which we draw major conclusions.

For our TWAS analysis, we did not perform use coloc SuSiE, and instead used FOCUS, for the reasons described in response to comment 1. We continue to report the default coloc results in the main text and Table 1 and note the conservativeness of coloc's single causal variant assumption in the main text (see response to comment 1).

3. Line 271-273 “Our study also identifies potential druggable genes, including genes located at chromosome 6p22.3 (TPMT and NHLRC1) that could promote human longevity.” I believe the wording here is somewhat too strong considering there was mixed evidence for causality of TPMT and NHLRC1.

Author response: We understand the reviewer's concern and have tempered our language to reflect the mixed evidence of causality for these genes. We now state: “Our study also identifies possible genetic drug targets, including genes located at chromosome 6p22.3 (*TPMT* and *NHLRC1*), that warrant further study for their involvement in human longevity.”

4. What does the column “In credible set” in Supplementary Data 7 refer to? If it is an indicator for which genes are included in the credible set, why is this value zero for some genes that have a posterior inclusion probability of one?

Author response: We thank the reviewer for bringing our attention to this error. We discussed this strange finding with Dr. Nicholas Mancuso, who developed the FOCUS fine-mapping method (PMID: 30926970). Dr. Mancuso informed us of an issue in a previous version of the FOCUS code pipeline that caused credible sets to be incorrectly calculated. We re-ran FOCUS using the latest version available on GitHub that has since resolved this issue. All genes with a PIP > 0.5 are included in the credible set at that locus, which Supplementary Data 7 now reflects.

5. Lines 157-159: “genes associated with all five traits had functions broadly related to insulin signaling, mitochondrial function and metabolism (Supplementary Data 11).” As far as I can tell there are no high confidence genes associated with all five traits. The authors mention the only genes associated with more than one trait are TPMT and NHLRC1 (which are associated with PhenoAge and IEAA).

Author response: You are correct. The intended meaning of this sentence was that genes associated with *each* of our five traits had functions related to insulin signaling, mitochondrial function, and metabolism. In other words, PhenoAge genes were implicated in the functions, as were IEAA genes, as were multivariate longevity genes, etc. To convey this meaning more precisely, we have replaced the quoted language with the following: “genes associated with each individual aging trait had functions broadly related to insulin signaling, mitochondrial function, cellular response to stress, and metabolism (Supplementary Data 11).”

6. Line 169: “Table 3” should be “Table 2” as the preceding sentence reports drug target MR results.

Author response: Thank you. We have corrected this error.

Reviewer #2 (Remarks to the Author):

The authors have made a careful and thorough revision addressing my concerns. I have a minor comment regarding the little converging signals among different clocks.

While the revision did recognize the issue, it is still unclear what implications these findings are on each clock. Given that five clocks have been extensively used for many studies, I hope the authors will make further efforts to discuss this topic in more detail.

Author response: We appreciate the reviewer's encouragement to further explore this question. We have expanded our discussion comparing the transcriptomic signatures of the epigenetic age acceleration outcomes, and we believe this has resulted in a more comprehensive paper.

Discussion: "At the transcriptomic level, there was one TWAS finding, *FLOT1*, shared between EAA (HannumAge) and multivariate longevity, which was only high confidence for HannumAge. Four TWAS associations were associated with multiple measures of EAA, including the high confidence associations of *TPMT* and *NHLRC1* with both IEAA and PhenoAge (Fig. 2). Functional analyses of our high confidence genes using GO revealed that, generally, genes implicated in each of our five aging phenotypes had different biological functions. The unique transcriptomic signatures of our four EAA phenotypes is particularly notable, yet not entirely unexpected, in the context of past research showing that different epigenetic clocks contain generally distinct CpG sites in distinct genomic regions.⁵⁸ We postulate that our four EAA measures may reflect unique epigenetic aging phenomena due to differences in training outcomes, tissues, and populations.⁵⁹ These epigenetic aging phenomena may capture various hallmarks of aging to different degrees. For example, PrismEXP implicated GrimAge (*SESNI*) and IEAA (*AKIRIN1*) genes in functions related to nutrient sensing, while a HannumAge gene (*KPNA4*) was implicated in the regulation of gene expression. Dysregulation of these processes are considered to play key roles in age-related physiological decline.^{60,61} However, we note that high confidence genes linked to each of our five aging outcomes were implicated in functions broadly related to insulin signaling, mitochondrial function, cellular response to stress, and metabolism. These biological domains may play fundamental roles in diverse aging-related phenotypes. Future *in vivo* and *in vitro* studies should attempt to better characterize the relationships of different epigenetic clocks and different biological process related to aging. Additionally, meta-clocks like the one described in Liu et al (2020),⁵⁸ which may capture fundamental aging processes and predict aging related health decline more effectively than individual component clocks, should be assessed as a potential aging biomarker."